# New Application for the Early Detection of Wound Infections Using a Near-Infrared Fluorescence Device and Forward-Looking Thermal Camera

**DOI:** 10.3390/diagnostics15172221

**Published:** 2025-09-01

**Authors:** Ha Jong Nam, Se Young Kim, Hwan Jun Choi

**Affiliations:** 1Department of Plastic and Reconstructive Surgery, Soonchunhyang University Gumi Hospital, Gumi 39371, Republic of Korea; namssi3@naver.com (H.J.N.); 111459@schmc.ac.kr (S.Y.K.); 2Department of Plastic and Reconstructive Surgery, Soonchunhyang University Cheonan Hospital, Cheonan 31151, Republic of Korea

**Keywords:** wound infections, near-infrared fluorescence, thermal imaging

## Abstract

**Background:** Timely and accurate identification of wound infections is essential for effective management, yet remains clinically challenging. This study evaluated the utility of a near-infrared autofluorescence imaging system (Fluobeam^®^, Fluoptics, Grenoble, France) and a thermal imaging system (FLIR^®^, Teledyne LLC, Thousand Oaks, CA, USA) for detecting bacterial and fungal infections in chronic wounds. Fluobeam^®^ enables real-time visualization of microbial autofluorescence without exogenous contrast agents, whereas FLIR^®^ detects localized thermal changes associated with infection-related inflammation. **Methods:** This retrospective clinical study included 33 patients with suspected wound infections. All patients underwent autofluorescence imaging using Fluobeam^®^ and concurrent thermal imaging with FLIR^®^. Imaging findings were compared with microbiological culture results, clinical signs of infection, and semi-quantitative microbial burdens. **Results:** Fluobeam^®^ achieved a sensitivity of 78.3% and specificity of 80.0% in detecting culture-positive infections. Fluorescence signal intensity correlated strongly with microbial burden (r = 0.76, *p* < 0.01) and clinical indicators, such as exudate, swelling, and malodor. Pathogens with high metabolic fluorescence, including *Pseudomonas aeruginosa* and *Candida* spp., were consistently identified. Representative cases demonstrate the utility of fluorescence imaging in guiding targeted debridement and enhancing intraoperative decision-making. **Conclusions:** Near-infrared autofluorescence imaging with Fluobeam^®^ and thermal imaging with FLIR^®^ offer complementary, noninvasive diagnostic insights into microbial burden and host inflammatory response. The combined use of these modalities may improve infection detection, support clinical decision-making, and enhance wound care outcomes.

## 1. Introduction

Wound infections remain a significant clinical challenge, particularly in patients with chronic wounds, diabetic ulcers, or burns. Infections delay healing, increase the risk of complications, and increase morbidity and healthcare costs. Traditional diagnostic methods, including clinical assessments and microbiological cultures, are often constrained by subjectivity, time delays, and potential false-negative results. Notably, clinical criteria alone frequently fail to detect wounds with high microbial burden [1,2,3]. This highlights the need for more accurate and real-time diagnostic tools to facilitate the early detection and timely management of wound infections [4,5]. Accordingly, the present work focuses on chronic wounds with clinically suspected infection managed in a surgical setting, and findings should be interpreted within this clinical context.

Recent advances in imaging technologies have expanded the diagnostic capabilities for wound care. Techniques such as spectral fingerprinting [6], fluorescence lifetime metabolic profiling [7], and real-time label-free optical detection [8,9] allow for the noninvasive visualization of microbial activity and tissue abnormalities. These innovations represent a shift toward optical biosensing for infection diagnostics [10,11]. In this manuscript, fluorescence refers to the optical phenomenon; autofluorescence denotes endogenous fluorescence emitted by microbial or tissue fluorophores without exogenous contrast agents, and microbial burden refers to the quantitative density of viable microorganisms measured from tissue culture and expressed as CFU/mL.

Recent advances in clinical wound imaging include point-of-care violet and visible excitation platforms that visualize porphyrin-related red emission and cyan emission associated with *Pseudomonas aeruginosa* in real time, label-free spectroscopic methods, and fluorescence-lifetime imaging capable of fingerprinting microbial metabolic state [2,7,8,9,12,13,14]. At the molecular level, *Pseudomonas aeruginosa* produces fluorescent siderophores such as pyoverdine, whereas *Staphylococcus aureus* accumulates porphyrins, which together explain these characteristic emission colors [15,16,17,18]. Because most reported bacterial fluorophores peak in the violet or visible range, extension to near-infrared excitation and detection as used in this study should be interpreted with caution and does not enable species identification without spectral decomposition.

The Fluobeam^®^ (Fluoptics Inc., Grenoble, France) is a near-infrared (NIR) fluorescence imaging system originally developed for intraoperative visualization [19]. It has been widely used in the clinical setting for lymphatic mapping, perfusion assessment during flap surgery, sentinel lymph node biopsy, and parathyroid gland identification [19,20,21,22,23]. Although its primary use involves perfusion assessment with indocyanine green, its application in wound care remains unexplored.

Although the Fluobeam^®^ system itself has not been previously reported for wound perfusion monitoring, other NIR fluorescence imaging systems have been successfully utilized to evaluate perfusion in chronic and diabetic wounds [24,25]. Based on this technical and conceptual similarity, we routinely employed the Fluobeam^®^ system to monitor perfusion across a variety of wound types. During these procedures, we unexpectedly observed distinct autofluorescence signals in the areas that were subsequently confirmed to be infected. This observation prompted an exploratory analysis of the ability of the device to detect infections via endogenous bacterial fluorescence, without the need for exogenous contrast agents.

Autofluorescence occurs when bacterial or fungal metabolites emit characteristic signals in response to specific wavelengths of light. Characteristic emissions in wound pathogens reflect endogenous fluorophores, including siderophores and porphyrins (e.g., pyoverdine in *P. aeruginosa* and porphyrins in *S. aureus*) [15,16,17,26]. Recent studies have demonstrated the utility of engineered dyes [27], bacterial probe systems [28], and fluorescence in situ hybridization for visualizing intact bacterial cells [29]. These methods confirm that fluorescence imaging allows rapid and safe point-of-care detection of infections without invasive sampling [9,10].

In parallel, thermal imaging technologies, such as the FLIR^®^ (Forward-Looking InfraRed, Wilsonville, OR, USA) system (Teledyne LLC, Thousand Oaks, CA, USA), offer additional diagnostic value by capturing subtle surface temperature elevations associated with underlying inflammation or infection. These systems have been proven effective in identifying subclinical changes in diabetic foot ulcers and postoperative wounds, often before the appearance of visible signs [30,31,32]. When combined, fluorescence and thermal imaging can provide complementary diagnostic information. While fluorescence detects the microbial burden, thermal imaging assesses the inflammatory response of the host.

The integration of these imaging modalities into wound care has shown benefits, such as improved diagnostic accuracy, reduced delays in treatment, and better guidance for individualized wound management [1,4,31]. Fluorescence imaging can help define appropriate debridement zones, optimize antimicrobial strategies, and enhance healing outcomes [5,31]. Some studies have reported sensitivity and specificity values exceeding 90% when combining thermal and fluorescence imaging for infection detection [18,31,33].

In this study, we introduce an off-label clinical application of the Fluobeam^®^ device for visualizing bacterial autofluorescence in infected wounds. We assessed the feasibility, diagnostic performance, and potential clinical value of this novel imaging strategy in routine wound care practice.

## 2. Materials and Methods

### 2.1. Study Design

This retrospective clinical study was conducted between May 2023 and April 2024 to evaluate the diagnostic performance of the Fluobeam^®^ device in detecting wound infections through autofluorescence imaging. The study was approved by the Institutional Review Board of the study hospital (IRB No. 2024-06-032), and written informed consent was obtained from all participants before enrollment.

### 2.2. Imaging System

The primary device used in this study was the Fluobeam^®^ (Fluobeam^®^ 800, Fluoptics, Grenoble, France), a handheld near-infrared fluorescence imaging system originally designed for intraoperative vascular and tissue perfusion assessment [19]. It emits excitation light at approximately 780 nm and detects signals in the near-infrared range at approximately 820 nm [19,20]. Although not initially intended for microbial detection, the device enables real-time, noninvasive visualization of autofluorescent signals that may correspond to bacterial or fungal colonization without the need for exogenous contrast agents [15,16]. Use of Fluobeam^®^ for bacterial/fungal autofluorescence in this study was off-label; no manufacturer-provided calibration for microbial detection was available. To standardize acquisition, imaging was performed in a controlled operating room (21–24 °C; 40–60% relative humidity) under consistent ambient lighting, using a fixed working distance (~20 cm), fixed exposure settings and detector gain, and a perpendicular camera orientation to the wound surface. Fluorescence intensity was quantified using the manufacturer’s software as mean grayscale values within user-defined regions of interest (ROIs); the underlying processing algorithms are proprietary and were not accessible. In the absence of device-level calibration for microbial detection, we enhanced comparability by applying ROI-based normalization to adjacent non-fluorescent reference skin under the fixed acquisition conditions described above.

### 2.3. Participants

A total of 43 patients with clinically suspected wound infections were enrolled in this study. Eligible participants were adults aged ≥18 years who presented with wounds exhibiting at least one clinical sign of infection, including erythema, swelling, warmth, purulent discharge, or malodor. Wound etiologies included diabetic foot ulcers, necrotic wounds, malignancy-associated wounds, and thermal injuries. Exclusion criteria comprised wounds unsuitable for imaging (e.g., because of excessive exudate or extensive tissue loss), known hypersensitivity to near-infrared light (e.g., photosensitivity disorders), severe systemic illness, or cognitive or neurological impairment precluding informed participation or protocol adherence. Wounds with excessive exudate were excluded to avoid optical attenuation and to ensure reproducible region-of-interest quantification under the standardized protocol.

### 2.4. Imaging Procedure

All imaging procedures were conducted in a controlled operating room maintained at a stable ambient temperature (21–24 °C) and relative humidity (40–60%) under consistent ambient lighting. Each imaging session followed a standardized sequence, beginning with fluorescence imaging using the Fluobeam^®^ device, followed by thermal imaging using the FLIR^®^ system. All images were acquired at a fixed distance of approximately 20 cm from the wound surface with fixed exposure settings and detector gain and a perpendicular camera orientation to the wound surface.

At the initial presentation, baseline fluorescence imaging was performed prior to wound cleaning to capture the wound in an unaltered state. Standard surgical debridement and saline irrigation were then performed to remove necrotic tissue and surface debris. A second round of fluorescence imaging was performed immediately following the same protocol using identical acquisition parameters.

Autofluorescent regions were identified during both sessions and documented using high-resolution still images and video recordings. The fluorescence intensity within the regions of interest was quantified using the manufacturer’s proprietary software; circular ROIs (~1 cm^2^) were placed on peak signal within the wound and on adjacent clinically uninvolved skin, and a relative fluorescence index (RFI; wound/reference ROI ratio) was computed to mitigate inter-frame variability. Fluorescence ROIs were co-registered between pre- and post-debridement images to maintain anatomic consistency. Multiple frames were captured per session to minimize motion or blur artifacts.

Thermal imaging was performed immediately after the fluorescence imaging under identical environmental and positional conditions. Skin emissivity was set to 0.98; ROIs (~1 cm^2^) were co-registered to the fluorescence maps, and temperatures were reported as ROI means. No blackbody-traceable thermal calibration was performed in this retrospective study; accordingly, small absolute temperature differences on the order of a few tenths of a degree Celsius (~0.3 °C) were treated as indeterminate, and interpretation prioritized spatial patterns (co-localization with fluorescence, perilesional extension, and—where available—serial change) over single absolute thresholds.

### 2.5. Microbiological Sampling and Pathogen Identification

After imaging, tissue samples were obtained from the autofluorescent regions for microbiological evaluation. Specimens were cultured on selective and differential media, including nutrient agar, MacConkey agar, and Sabouraud dextrose agar, to isolate aerobic and facultative anaerobic bacteria (e.g., *P. aeruginosa*, *Escherichia coli*, *Klebsiella pneumoniae*) and fungal pathogens (e.g., *Candida* species). For quantitative analysis, the tissue samples were weighed, homogenized in sterile phosphate-buffered saline, and serially diluted. The diluted suspensions were plated onto appropriate media, incubated, and analyzed for colony-forming units (CFU). The microbial burden was expressed as CFU per milliliter of homogenate (CFU/mL). Microbial identification was conducted using standard laboratory techniques, including Gram staining, biochemical assays, and polymerase chain reaction, as necessary, to confirm the species-level classification. Microbial burden was analyzed as a continuous variable (CFU/mL); semi-quantitative CFU categories were not assigned. Routine anaerobic culture was not performed in this retrospective cohort, which may underestimate anaerobic pathogens in chronic wounds.

### 2.6. Statistical Analysis

Quantitative data from fluorescence imaging and microbiological cultures were analyzed to evaluate the association between the autofluorescence signal intensity and the presence of microbial infection. Fluorescence intensity was summarized using RFI, and microbial burden was treated as a continuous variable (CFU/mL). Descriptive statistics (mean, median, and standard deviation) were used to summarize the patient demographics and fluorescence measurements. Comparative analyses between fluorescence–positive- and negative-wounds were conducted using the chi-square test or Fisher’s exact test, as appropriate. Pearson’s correlation coefficient (r) was used to assess the linear relationship between the fluorescence intensity (RFI) and quantified microbial burden (CFU/mL). Diagnostic performance metrics (sensitivity, specificity, positive and negative predictive values) were calculated by comparing fluorescence results against culture as the reference standard. For descriptive reporting in Results, fluorescence intensity tiers (low, moderate, high) were used to aid presentation; all inferential analyses used continuous RFI and CFU/mL values. Statistical significance was set at *p* < 0.05. All statistical analyses were performed using SPSS Statistics software (version 30.0.0; IBM Corp., Armonk, NY, USA).

## 3. Results

### 3.1. Patient Demographics

A total of 33 patients (24 men and 9 women) were included in the study. The mean age of the patients was 58 years (range: 42–78 years). Diabetic foot ulcers were the most common wound type (13 patients, 39.4%), followed by necrotic wounds (8, 24.2%), cancer-related wounds (*n* = 6, 18.2%), and burns (*n* = 6, 18.2%) (Table 1). All wounds were chronic and underwent standard initial management before imaging, including debridement and dressing application, in accordance with the institutional protocol. Given the small and uneven subgroup sizes across wound etiologies, no inferential comparisons by wound type were performed.

### 3.2. Diagnostic Accuracy of Fluorescence Imaging

Of the 33 wounds analyzed, 69.7% (23/33) had microbiologically confirmed infections, whereas 30.3% (10/33) yielded negative cultures. Fluobeam^®^ imaging demonstrated positive bacterial fluorescence in 20 wounds and negative fluorescence in 13. As summarized in Table 2, the device correctly identified 18 of 23 infected wounds (true positives) and failed to detect 5 infected wounds (false negatives). It also correctly excluded infections in 8 of 10 culture-negative wounds (true negatives), with two false positives. These results corresponded to a sensitivity of 78.3% (95% CI, 58.1–90.3%) and a specificity of 80.0% (95% CI, 49.0–94.3%). The positive and negative predictive values were 90.0% and 61.5%, respectively. The overall diagnostic accuracy of the fluorescence imaging modality in this cohort was 78.8%. These data suggest that near-infrared fluorescence imaging is a clinically useful adjunct to standard infection assessment in chronic wounds.

### 3.3. Fluorescence Signal and Microbial Burden

A strong positive correlation was observed between the fluorescence signal intensity and microbial burden (Pearson’s r = 0.76, *p* < 0.01), indicating that wounds emitting stronger fluorescent signals tended to harbor higher concentrations of bacteria. When stratified into qualitative intensity tiers (low, moderate, and high), the fluorescence intensity exhibited a gradational association with microbial burden and clinical severity. As shown in Table 3, wounds with high-intensity fluorescence demonstrated the greatest microbial burden (mean 9.2 × 10^6^ CFU/mL) and were clinically categorized as severe infections. Moderate-intensity fluorescence was associated with intermediate bacterial counts and moderate clinical signs, whereas low-intensity fluorescence corresponded to relatively mild infections with lower microbial burdens (mean 2.9 × 10^4^ CFU/mL). These findings support the potential utility of fluorescence signal gradation as a semi-quantitative indicator of infection severity.

### 3.4. Association with Clinical Signs of Infection

The fluorescence signal intensity was positively correlated with several classical clinical signs of infection. Among them, the amount of exudate demonstrated the strongest correlation (r = 0.72; *p* = 0.004), followed by swelling (r = 0.68; *p* = 0.006), and foul odor (r = 0.68; *p* = 0.008). In contrast, the correlations between induration (r = 0.65, *p* = 0.081) and pain (r = 0.59, *p* = 0.137) were not statistically significant (Table 4). These correlations were calculated without covariate adjustment in this preliminary cohort and may be confounded by wound size, chronicity, or comorbid conditions; accordingly, they should be viewed as hypothesis-generating.

### 3.5. Thermal Imaging Findings

As an adjunct assessment, the wound surface temperature was evaluated using thermal imaging with a FLIR^®^ camera. All thermal images were acquired immediately after Fluobeam^®^ imaging under standardized conditions for both fluorescence-positive and fluorescence-negative wounds.

Wounds exhibiting positive fluorescence signals showed a slightly elevated mean surface temperature (36.8 °C ± 0.3) compared to fluorescence-negative wounds (36.5 °C ± 0.2) (Table 5). However, the mean absolute temperature difference was modest (~0.3 °C) and, in the absence of device-level calibration, should be interpreted cautiously; emphasis was placed on spatial concordance and perilesional spread rather than the magnitude of the delta. The localized hyperthermia was spatially consistent with areas of fluorescence positivity, suggesting concurrent inflammatory activity or infection.

In several cases, thermographic signals extended beyond the borders of the fluorescence-positive zones. This peripheral thermal elevation may reflect perilesional inflammatory responses that have not yet been associated with detectable bacterial colonization. In this cohort, thermal–fluorescence relationships were assessed qualitatively; the frequency and magnitude of perilesional thermal extension will be quantified prospectively using pre-specified spatial overlap metrics (e.g., Jaccard index) and a thermal-overreach ratio. These observations highlight the complementary diagnostic utility of dual imaging. Although Fluobeam^®^ effectively localizes microbial presence based on autofluorescence, FLIR^®^ imaging provides additional information on the spatial extent of inflammation, potentially offering earlier insights into surrounding tissue responses.

### 3.6. Microbiological Findings

Culture analysis identified a variety of bacterial and fungal pathogens in the infected wounds. Table 6 summarizes the fluorescence detection rates for each organism.

The fluorescence imaging system demonstrated 100% detection of *P. aeruginosa* (6/6) and fungal isolates, such as *Candida* spp. (4/4), likely because of the inherent autofluorescent properties of these organisms. The detection of *S. aureus* was slightly lower, with six of the eight culture-positive wounds yielding positive fluorescence signals (75.0%).

For gram-negative enteric bacteria, the detection rate was moderate; Klebsiella pneumoniae and Escherichia coli were detected in one of the two culture-positive wounds (50.0%). Notably, Enterococcus faecalis was not detected in any culture-positive case (0%). Because several taxa were represented by very small numbers (e.g., Enterococcus faecalis *n* = 1), species-level detection proportions are reported descriptively without inferential claims.

### 3.7. Species-Specific Detection Limits and Autofluorescent Properties

Table 7 summarizes the pathogens isolated in this cohort, their detection rates within the study, and autofluorescent metabolites and properties reported in the literature under the 780 nm excitation and 820 nm emission detection settings used. Detection rates were categorized as High (≥90%), Moderate (50–89%), and Low (<50%), based on the proportion of fluorescence-positive results among culture-confirmed wounds for each species. These detection rates are descriptive and cohort-specific, and do not represent analytical limits of detection or comprehensive device performance metrics. Organisms with well-characterized endogenous fluorophores, such as *P. aeruginosa* and *S. aureus*, demonstrated higher detection rates. In contrast, Candida spp. showed high detectability in this cohort, but their endogenous fluorophores remain poorly characterized; current evidence suggests the presence of putative porphyrin-related fluorophores, though this requires further investigation. Gram-positive cocci exhibited lower detection rates. Anaerobic species were not systematically evaluated and are not represented.

#### 3.7.1. Case 1

A 75-year-old male with diabetes mellitus and hypertension presented with a necrotic ulcer on the left plantar foot that developed after inotropic therapy. Surgical debridement was performed, and the defect was covered with a skin graft (Figure 1A). Near-infrared fluorescence imaging demonstrates distinct autofluorescence at the wound margins without the use of contrast agents. No signal was detected in the central grafted area or in the adjacent healthy skin treated with povidone-iodine. *P. aeruginosa* was isolated from tissue cultures obtained from the autofluorescent region. Concurrent thermal imaging revealed localized hyperthermia with a peak surface temperature of 33.5 °C, consistent with inflammation (Figure 1B). The spatial correlation between the autofluorescence signal and thermal elevation matched the area of bacterial colonization, as confirmed by microbiological analysis (Figure 1C,D).

#### 3.7.2. Case 2

A 48-year-old male presented with a Marjolin ulcer in the sacral region characterized by chronic exudate and wet debris. After surgical debridement, a skin graft was applied to cover the necrotic tissue (Figure 2A). Autofluorescence imaging using the Fluobeam^®^ system revealed a central fluorescence signal within the ulcer bed, without the use of contrast agents. No fluorescence was detected in the surrounding healthy tissues. A tissue culture of the autofluorescent region confirmed infection with *P. aeruginosa*. Thermal imaging demonstrated localized hyperthermia at the site of fluorescence (peak temperature: 34.6 °C), supporting the presence of underlying infection (Figure 2B–D).

#### 3.7.3. Case 3

A 55-year-old male with diabetes mellitus and hypertension presented with a necrotic ulcer on the right foot. After surgical debridement, a full-thickness skin graft was applied to cover the defect (Figure 3A). Autofluorescence imaging performed without contrast agents revealed a distinct peripheral signal localized along the wound margin, whereas no fluorescence was observed in the centrally grafted area. Tissue cultures obtained from the fluorescent regions confirmed the presence of S. aureus. Thermal imaging demonstrated a localized temperature elevation (peak: 33.6 °C) at the wound margin, corresponding spatially to the autofluorescent area and the site of confirmed infection (Figure 3B–D).

Real-time autofluorescence imaging was subsequently used to guide the selective debridement. As shown in Figure 4A, the fluorescence-positive areas were identified and marked along the graft margin using a surgical dye. These zones were then selectively excised, and tissue cultures were obtained from both the fluorescent and non-fluorescent regions. Post-debridement imaging (Figure 4B) demonstrated the complete resolution of fluorescence at the wound site, whereas the excised tissue retained a positive signal. The persistence of fluorescence in the debrided specimen supports its role as an active infection focus.

#### 3.7.4. Case 4

A 59-year-old female with a history of hypertension presented with dystrophic changes in the fingernails of the left hand that were clinically suggestive of onychomycosis. The contralateral (right) hand appeared grossly normal and was used as the internal control (Figure 5A). Near-infrared autofluorescence imaging using the Fluobeam^®^ device revealed strong fluorescence signals in the second to fourth fingernails of the left hand, without the administration of any contrast agent. No fluorescence was observed in the right hand of the mice. Fungal cultures obtained from the fluorescent zones confirmed infection with Trichophyton rubrum. Complementary thermal imaging showed localized temperature elevation over the infected fingernails, whereas the overall hand temperature remained symmetrical (Figure 5B–D). These findings underscore the ability of fluorescence imaging to clearly differentiate infected from non-infected tissues and support its diagnostic utility in suspected cases of onychomycosis.

## 4. Discussion

This study investigated a novel clinical application of Fluobeam^®^, a near-infrared autofluorescence imaging device originally designed for intraoperative perfusion assessment, lymphatic mapping, and parathyroid gland identification [19,20,21,22,23]. In this study, we repurposed Fluobeam^®^ for detecting bacterial and fungal infections in chronic wounds by visualizing endogenous autofluorescence without the use of exogenous contrast agents. Our findings suggest that Fluobeam^®^ can serve as a noninvasive, real-time, point-of-care diagnostic tool capable of localizing microbial foci with clinically meaningful accuracy [9,10]. Taken together, our findings support the feasibility of Fluobeam^®^ as a noninvasive, real-time, point-of-care adjunct that can localize microbial foci, rather than establishing superiority over existing methods. This study should be interpreted as a preliminary, single-center, retrospective analysis with a small sample size (*n* = 33), which limits precision and external validity. Because the cohort was enriched for clinically suspected infections and we excluded wounds with excessive exudate, the reported performance may be overestimated relative to unselected wound populations.

Importantly, this analysis focused not on profiling the entire wound microbiota, but rather on identifying overt or clinically suspected infections in chronic wounds, contexts in which early detection can critically influence outcomes [1,4,5]. Fluorescence signal intensity was found to correlate with microbial burden, classical clinical signs of infection, and localized thermal changes, supporting its potential utility as an adjunct imaging modality for routine wound surveillance [2,16,32,34].

The fluorescence detection capability of Fluobeam^®^ relies on the intrinsic autofluorescence properties of certain microorganisms. Gram-negative bacteria, such as *P. aeruginosa*, produce fluorescent siderophores, notably pyoverdine, which emit cyan-green fluorescence under near-infrared or violet light excitation [15,18]. Cyan fluorescence has been clinically validated as a reliable signal for the identification of *P. aeruginosa* in infected wounds [13].

In contrast, many gram-positive species, including *S. aureus*, produce porphyrin-based metabolites that fluoresce red [16,18]. Fungi, such as *Candida* spp. and *Trichophyton* spp., have also been observed to generate detectable autofluorescence, although the biochemical origin of these signals remains unclear [18,35]. Although Fluobeam^®^ does not enable species-level resolution, it effectively highlights the regions of microbial metabolic activity. Strong fluorescence signals were most frequently observed in infections caused by *Pseudomonas* and *Candida*, consistent with the known biosynthesis of highly fluorescent metabolites by these organisms [15,18].

Rather than functioning solely as a binary diagnostic tool, Fluobeam^®^ demonstrated the capacity to provide semi-quantitative information regarding microbial burden. In our study, the fluorescence signal intensity closely reflected the bacterial density within the wound, supporting its use as a dynamic marker of infection severity [2,3,16]. As shown in Table 2, the device achieved a sensitivity of 78.3% (18/23) and a specificity of 80.0% (8/10) for detecting culture-positive infections, suggesting a robust diagnostic performance with an acceptable false-positive rate [5,36]. However, our sensitivity (78.3%) and specificity (80.0%) are below the >90% reported in some prior studies. Several factors may account for this difference: the preliminary, small single-center cohort; a case mix dominated by chronic, previously debrided wounds; the off-label use of a device without species-specific bacterial-detection calibration; reliance on surface signals that can miss deep-seated or biofilm-associated infections; and lack of routine anaerobic culture in all cases. Consistent with these factors, the negative predictive value was modest (61.5%), indicating that negative fluorescence does not rule out infection. Accordingly, for fluorescence-negative wounds with persistent clinical suspicion—particularly when non-fluorogenic pathogens (e.g., Enterococcus spp. or anaerobes) are plausible—we recommend confirmatory testing (tissue culture including anaerobic coverage and, when available, molecular assays such as PCR).

The high positive predictive value (90.0%) indicates that a positive fluorescence signal is strongly associated with a microbiologically confirmed infection. Conversely, the lower negative predictive value (61.5%) underscores a critical limitation: the absence of fluorescence does not reliably exclude infection, particularly in cases of deep-tissue, metabolically dormant, or biofilm-associated pathogens that may evade surface-based detection [1,4,12].

When the fluorescence intensity was stratified into low, moderate, and high categories, a clear correlation was observed between increasing microbial burden and clinical severity. As detailed in Table 3, high-intensity signals corresponded to a mean microbial burden of 9.2 × 10^6^ CFU/mL and were associated with severe infection, whereas low-intensity signals were observed in wounds with a mean load of 2.9 × 10^4^ CFU/mL, reflecting milder clinical presentations. This gradation reinforces the concept that autofluorescence imaging provides more than a binary output and can serve as a semi-quantitative adjunct for clinical decision-making [2,9,16].

Such signal stratification may assist clinicians in prioritizing focal debridement or adjusting antimicrobial strategies based on microbial density [5,12]. Moreover, fluorescence intensity was significantly correlated with key clinical indicators of infection. As shown in Table 4, strong positive correlations were noted with the exudate amount (r = 0.72, *p* = 0.004), swelling (r = 0.68, *p* = 0.006), and foul odor (r = 0.68, *p* = 0.008). These findings support the use of fluorescence imaging as an objective proxy for traditional clinical signs that are often subjective and inconsistently interpreted [1,3,4]. These correlations were calculated without covariate adjustment in this preliminary cohort and should be regarded as hypothesis-generating. This advantage is particularly relevant in patients with impaired sensation (e.g., diabetic neuropathy), chronic inflammation, or darker skin phototypes, where visual inspection may be unreliable [4,37].

A notable strength of this study is its dual-modality approach, which integrates FLIR^®^ thermal imaging with near-infrared autofluorescence. Although thermal imaging has been independently explored in previous reports, its concurrent application with fluorescence imaging remains uncommon in clinical wound assessment [32,36,37,38]. In the present study, wounds with positive fluorescence signals consistently demonstrated modest increases in surface temperature. As shown in Table 5, the mean temperature of fluorescence-positive wounds was 36.8 °C, compared to 36.5 °C in fluorescence-negative wounds—a mean differential of approximately 0.3 °C [32,38]. Given the absence of device-level thermal calibration and the small absolute delta (~0.3 °C), these differences should be interpreted cautiously; we prioritized spatial concordance with fluorescence and perilesional spread over reliance on absolute temperature thresholds [32,37,38].

These thermal elevations were typically observed in regions corresponding to the fluorescence signal and, in some cases, extended slightly beyond the fluorescent margins. This observation implies that FLIR^®^ thermal imaging may capture adjacent inflammatory zones that are not yet colonized by a detectable microbial burden [30,37]. Such perilesional thermal changes likely reflect early immune responses and may help to define broader areas of concern during infection surveillance or surgical debridement. Accordingly, in this cohort thermal imaging served as an adjunctive, pattern-based tool rather than a stand-alone diagnostic test, with clinical utility contingent on co-localization and spatial extent rather than on a fixed temperature cut-off [32,36,37,38].

Clinical implications of small thermal differences. The average thermal contrast observed in this cohort was small (~0.3 °C). In patients with vascular compromise (e.g., peripheral arterial disease) or under cool ambient conditions, baseline skin temperature and thermal gradients may be blunted, further limiting the diagnostic value of small absolute deltas. Therefore, in this study thermal imaging primarily served to map spatial patterns (co-localization with fluorescence and perilesional extension) rather than to apply absolute thresholds. Future work will incorporate blackbody-traceable calibration and repeated-measures assessments to define device-specific reproducibility and clinically meaningful temperature differentials [32,37,38].

In the present study, *P. aeruginosa* and fungal pathogens, such as *Candida* species, were detected with 100% sensitivity using autofluorescence imaging, whereas *S. aureus* was detected in 75% of culture-positive wounds. In contrast, *E. faecalis* was not detected in any case. These results, presented in Table 6, reflect the varying detection rates among pathogen species and align well with known metabolic properties. For instance, *P. aeruginosa* produces fluorescent siderophores, such as pyoverdine, which emit strong cyan signals under near-infrared excitation [15,18]. Similarly, *S. aureus* and *Candida* spp. generate porphyrin-based fluorophores that fluoresce red [16,18,35]. Conversely, low-metabolizing, or anaerobic organisms, such as *E. faecalis*, may lack the capacity to generate detectable levels of fluorophores [34].

These findings indicate that Fluobeam^®^ is highly effective in identifying pathogens capable of emitting autofluorescence, but may have limited sensitivity for organisms that lack such metabolic activity. Therefore, the device should be considered species-sensitive and capable of detecting a subset of clinically significant pathogens based on their unique fluorescence signatures, rather than functioning as a universal detection tool for all microbial species.

The clinical applicability of Fluobeam^®^ was further illustrated through four representative cases, each demonstrating distinct diagnostic or surgical advantages. These cases are illustrative use scenarios (targeted debridement and sampling localization) rather than evidence of outcome benefit or clinical efficacy.

In Case 1, a marginal fluorescence signal precisely delineated a *P. aeruginosa* infection at the periphery of the grafted plantar foot ulcer. No fluorescence was observed in the surrounding tissue that had been pretreated with the antiseptic, reinforcing the spatial specificity of the device in identifying infected zones. In Case 2, a centrally located wound within a Marjolin ulcer exhibited a strong fluorescence signal that co-localized with localized thermal elevation and a culture-positive result. This case highlights the value of fluorescence imaging in identifying superimposed infections, even within malignancy-associated wounds, where microbial colonization may complicate surgical planning and reconstruction. In Case 3, intraoperative fluorescence imaging detected a marginal *S. aureus* infection at the edge of a split-thickness skin graft. Real-time visualization enabled targeted excision of fluorescence-positive areas, allowing for the preservation of healthy graft tissue and demonstrating the potential of Fluobeam^®^ to refine surgical margins and enhance intraoperative decision-making. In Case 4, localized autofluorescence signals were detected in the second to fourth fingernails of the left hand of a patient with suspected onychomycosis. The right hand, which showed no fluorescence, served as the internal negative control. Fungal cultures confirmed *Trichophyton rubrum* in the autofluorescent nails. This case extends the diagnostic relevance of Fluobeam^®^ beyond wound surveillance, illustrating its capability to detect superficial cutaneous fungal infections.

Figure 6 illustrates longitudinal outcomes for three cases. Panel A shows the 3-month status of Case 1 with partial epithelialization and a small residual slough at the same peripheral margin that corresponded to the baseline fluorescence-positive hotspot, indicated by a yellow dashed ellipse. During follow-up this region underwent targeted debridement and then progressed to complete epithelialization. Panels B and C show the 12-month outcomes of Cases 2 and 3 with durable epithelialization. These images are presented as illustrative use scenarios rather than evidence of outcome benefit, and standardized serial fluorescence and thermal acquisitions were not available in this retrospective cohort.

The findings of this study support the integration of fluorescence imaging as an adjunct diagnostic tool in wound management. When Fluobeam^®^ and FLIR^®^ thermal imaging are combined, these modalities create a complementary diagnostic framework. Fluobeam^®^ enables real-time visualization of microbial colonization, whereas FLIR^®^ captures inflammatory changes in the adjacent tissue. This combination enhances intraoperative decision-making, guides targeted debridement, and reduces reliance on delayed laboratory-based culture diagnostics (Table 2 and Table 5) [1,5,32].

Although this was a preliminary investigation, the use of a combination autofluorescence and thermal imaging represents a novel approach. Previous studies have rarely used both modalities simultaneously to assess wound infections [36,38]. Our findings suggest that integrating these techniques yields a more comprehensive assessment of both microbial activity and host responses, particularly in wounds with subtle or ambiguous clinical signs [4,32].

It is important to note that this study focused on infection sites with a clinically apparent microbial burden. Our intention was not to detect the full range of potential wound pathogens. As shown in Table 6, the detection capability of Fluobeam^®^ varied by species. Organisms such as *P. aeruginosa* and *Candida* spp. were consistently identified, whereas others such as *E. faecalis* were not detected. This outcome reflects the fluorescence-dependent metabolic characteristics of each pathogen, and emphasizes the need for species-specific interpretations when using this imaging modality [18,34].

These species-level patterns are consistent with fluorescence biology reported for wound pathogens: organisms that biosynthesize strong endogenous fluorophores, such as pyoverdine in *P. aeruginosa* and porphyrins in *S. aureus*—tend to be more readily detected, whereas taxa with weak or absent fluorophore production (e.g., gram-positive cocci such as *Enterococcus* spp. and many anaerobes) show poorer detectability under our 780-nm excitation/820-nm detection protocol. Because most documented bacterial fluorophores exhibit excitation/emission peaks in the violet or visible range, extension to near-infrared excitation and detection should be interpreted with caution [14,15,16,17,18,28]. Moreover, the device does not perform spectral decomposition, therefore, fluorescence alone does not enable species identification. Accordingly, Table 7 is provided as an interpretive aid that links our observed detection proportions with literature-based autofluorescent metabolites and typical visible-range characteristics rather than as a species-identification key.

From a clinical standpoint, a negative fluorescence finding should not be taken to exclude infection when low- or non-fluorogenic organisms are plausible (e.g., *Enterococcus* spp. or anaerobes); confirmatory testing with tissue sampling, including anaerobic culture—and, where available, molecular assays (e.g., PCR) is recommended. Conversely, fluorescence-positive areas can be used to target debridement and to localize culture sampling to the highest-yield zone, potentially improving microbiological confirmation and procedural efficiency [2,13,14,27].

Positioning the dual modality among adjunct tools. To place our dual-modality approach in context, we added a concise comparison of adjunct tools (Table 8). Fluobeam^®^ + FLIR^®^ provides rapid bedside mapping of microbial foci and perilesional spread but is off-label for bacterial detection, offers organism-dependent detectability, lacks spectral identification, and small absolute thermal deltas (~0.3 °C) warrant cautious interpretation without device-level calibration [2,32,37,38]. MolecuLight^®^ is a purpose-built, portable point-of-care fluorescence platform with robust clinical evidence for bacterial-burden mapping and guidance of debridement/culture, although visibility depends on porphyrin/cyan fluorophore production and ambient-light constraints; species-level identification is not provided [12,14]. PCR delivers high analytic sensitivity/specificity and can detect targets/resistance but lacks viability confirmation, spatial mapping, and rapid turnaround compared with imaging [4]. Culture remains the clinical reference for organism identification and antimicrobial susceptibility, albeit with slower turnaround and potential false-negatives in fastidious/biofilm organisms [4]. We did not conduct head-to-head performance testing against these tools in this study; comparative accuracy and decision-impact will be evaluated prospectively. Taken together, Table 8 highlights complementary roles: imaging rapidly maps where to sample and treat, whereas laboratory tests determine what is present and how to target therapy.

A prospective study is underway to expand on these findings. This extended analysis aims to evaluate the diagnostic performance of fluorescence imaging across a wider microbial spectrum. Target organisms include additional gram-positive and gram-negative bacteria, anaerobes, and less common fungal species. A reference panel is being developed for inclusion as Table 9 in future reports.

In addition to its clinical relevance, fluorescence-based bacterial detection offers scalability for broader diagnostic applications. The Fluobeam^®^ system is portable, noninvasive, and provides immediate visual feedback without the need for specialized reagents or prolonged culture times [1,5,9]. These features suggest a strong potential for application in outpatient clinics, home care settings, and resource-limited environments, where conventional diagnostics may be inaccessible or delayed. Previous studies have highlighted the practical advantages of label-free point-of-care imaging systems for wound surveillance, supporting their utility in frontline infection control efforts [2,14].

As efforts continue toward miniaturized and reagent-free diagnostics, fluorescence imaging remains a practical and scalable adjunct for real-time infection monitoring. Parallel research is also underway to investigate its utility in detecting clinically important drug-resistant organisms, such as methicillin-resistant *S. aureus* and carbapenem-resistant *Enterobacteriaceae*. These pathogens present significant challenges in wound care and require early identification to guide antimicrobial stewardship. Future studies should evaluate whether fluorescence signal characteristics, including intensity, persistence, and spectral profile, can serve as surrogate markers of microbial virulence, resistance phenotypes, or biofilm formation [7,30]. The results of this study will be reported separately.

This study had several limitations. First, it was a preliminary, single-center, retrospective study with a small sample size (*n* = 33), which limits precision and generalizability across heterogeneous wound types and microbial spectra. Inclusion was restricted to clinically suspected infections; subclinical infections and non-infectious inflammatory conditions were not systematically included, which may bias case mix and overestimate diagnostic performance relative to unselected populations. Second, to avoid optical attenuation and ensure reproducible ROI placement, we excluded wounds with excessive exudate; together with enrichment for clinically apparent infection, this may introduce selection bias. Third, although fluorescence intensity correlated with infection severity, the device lacks species-level resolution; species-level detection proportions were underpowered for several taxa (e.g., rare isolates) and are reported descriptively without inferential claims. Fourth, thermal findings were based on modest surface temperature differences (~0.3 °C) acquired without blackbody-traceable device calibration; such small deltas may fall within physiologic or measurement variability and can be influenced by environmental or vascular factors (e.g., peripheral vascular disease). We did not stratify performance by wound location (hair-bearing vs. glabrous skin), skin phototype, or operator experience, which may affect signal visibility and ROI placement, and we did not perform standalone-versus-combined receiver operating characteristic analyses, so the incremental benefit of the dual modality was not quantified. In addition, Fluobeam^®^ was used off-label for bacterial/fungal autofluorescence without microorganism-specific calibration; thermal–fluorescence co-localization was assessed qualitatively without pre-specified spatial metrics; and correlations between fluorescence and clinical signs were unadjusted and susceptible to confounding (e.g., wound size, chronicity, comorbidities). Finally, culture-based reference standards have limitations, particularly for fastidious or biofilm-associated organisms, and routine anaerobic culture was not performed in all cases. To mitigate these limitations, we plan a prospective, adequately powered, multicenter study (target > 100 participants) spanning diverse wound etiologies (burns, diabetic foot ulcers, malignancy-associated wounds), with consecutive enrollment (including subclinical infections and non-infectious mimics), pre-specified endpoints and subgroup powering, blinded image interpretation, expanded microbiology including anaerobes and molecular assays, and device calibration and repeatability assessments for fluorescence and thermal imaging. Future work will include serial, calibrated Fluobeam^®^/FLIR^®^ acquisitions during long-term follow-up with blinded readers to test reproducibility and clinical relevance of longitudinal signal changes.

Despite these limitations, the present study highlights the clinical promise of near-infrared autofluorescence imaging with Fluobeam^®^ as a noninvasive, point-of-care tool for detecting bacterial and fungal wound infections. Fluorescence intensity was correlated with microbial burden, culture results, and objective clinical signs, supporting its potential as a semi-quantitative diagnostic adjunct. When combined with FLIR^®^ thermal imaging, which captures periwound temperature changes indicative of inflammatory activity, the dual-modality approach provides complementary information on microbial colonization and host response. With broader validation in larger and more diverse patient populations, this integrated imaging strategy may improve diagnostic accuracy, guide targeted interventions, and enhance infection control in wound care practice.

## 5. Conclusions

This study demonstrates a novel off-label use of the Fluobeam^®^ near-infrared fluorescence system for detecting bacterial and fungal infections in chronic wounds with clinically suspected infection through autofluorescence imaging. The fluorescence signal showed meaningful correlations with microbial burden, clinical signs, and thermal imaging patterns, indicating its potential as a real-time, noninvasive diagnostic adjunct for localizing microbial foci rather than species identification. Combined use with FLIR^®^ thermal imaging provided complementary information on host response and spatial extent of infection severity. These findings support the integration of fluorescence-based imaging in routine wound care, while recognizing species-dependent detectability and modest absolute thermal differences. However, prospective, adequately powered studies are required to validate its performance in broader clinical settings and pathogen spectra.

## Figures and Tables

**Figure 1 diagnostics-15-02221-f001:**
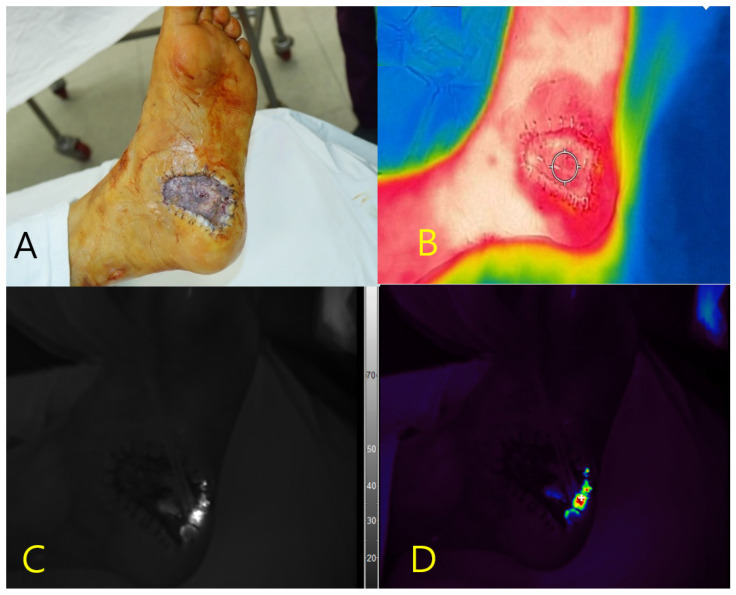
Multimodal imaging of a necrotic foot ulcer in a 75-year-old male patient. (**A**) Clinical photograph of the left plantar foot after debridement and skin grafting. (**B**) Thermal image demonstrating localized hyperthermia (peak temperature: 33.5 °C) at the wound margin. (**C**) Near-infrared fluorescence image showing autofluorescence at the peripheral wound area without contrast agent administration. (**D**) Pseudocolor overlay of the fluorescence signal corresponding to the region colonized by *Pseudomonas aeruginosa*, as confirmed by tissue culture. No fluorescence was observed in the adjacent skin pretreated with povidone-iodine.

**Figure 2 diagnostics-15-02221-f002:**
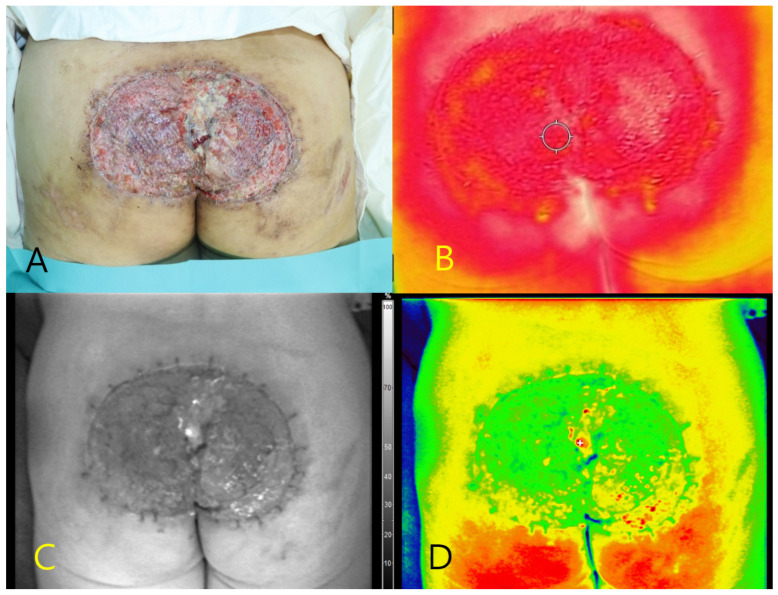
Multimodal imaging of a sacral Marjolin ulcer in a 48-year-old male patient. (**A**) Clinical photograph showing the ulcerative wound in the sacral region after debridement. (**B**) Thermal image demonstrating elevated surface temperature (peak: 34.6 °C) at the central portion of the ulcer. (**C**) Near-infrared fluorescence image showing autofluorescence in the central wound bed without contrast agent administration. (**D**) Pseudocolor overlay highlighting the autofluorescent region corresponding to the area where *Pseudomonas aeruginosa* was confirmed via tissue culture.

**Figure 3 diagnostics-15-02221-f003:**
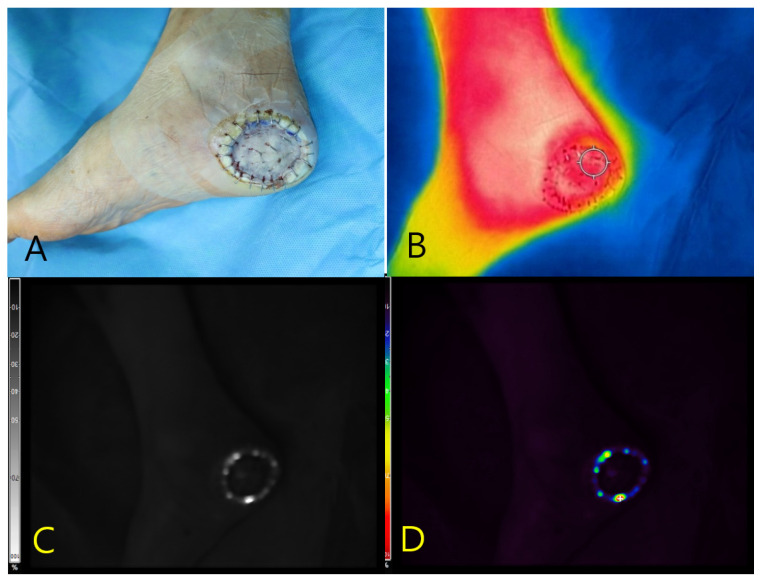
Multimodal imaging of a necrotic foot ulcer in a 55-year-old male patient. (**A**) Clinical photograph showing a circular full-thickness skin graft applied to the right plantar foot following surgical debridement. (**B**) Thermal image demonstrating marginal temperature elevation (peak: 33.6 °C) localized along the wound periphery. (**C**) Near-infrared fluorescence image revealing autofluorescence confined to the graft margin, obtained without the use of contrast agents. (**D**) Pseudocolor overlay localizing the fluorescence signal to the area subsequently confirmed to harbor *Staphylococcus aureus* on tissue culture.

**Figure 4 diagnostics-15-02221-f004:**
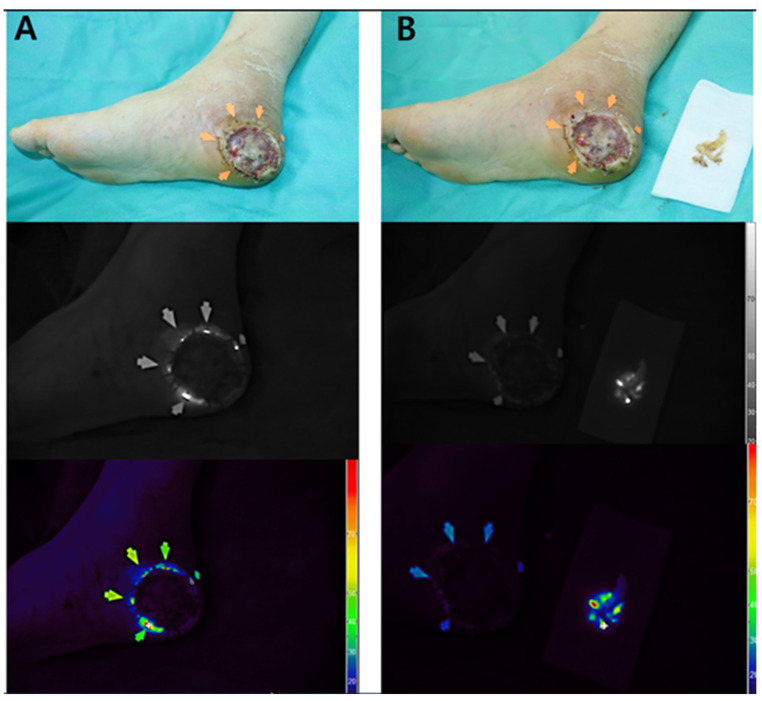
Intraoperative application of Fluobeam^®^ imaging for targeted debridement (Case 3). (**A**) Fluorescence-positive zones (arrows) delineated along the graft margin and marked with surgical dye to guide selective excision. (**B**) Post-debridement view showing complete disappearance of fluorescence at the wound site, while the excised tissue retains a strong autofluorescent signal. This finding supports the excised region as the microbiologically active infection focus.

**Figure 5 diagnostics-15-02221-f005:**
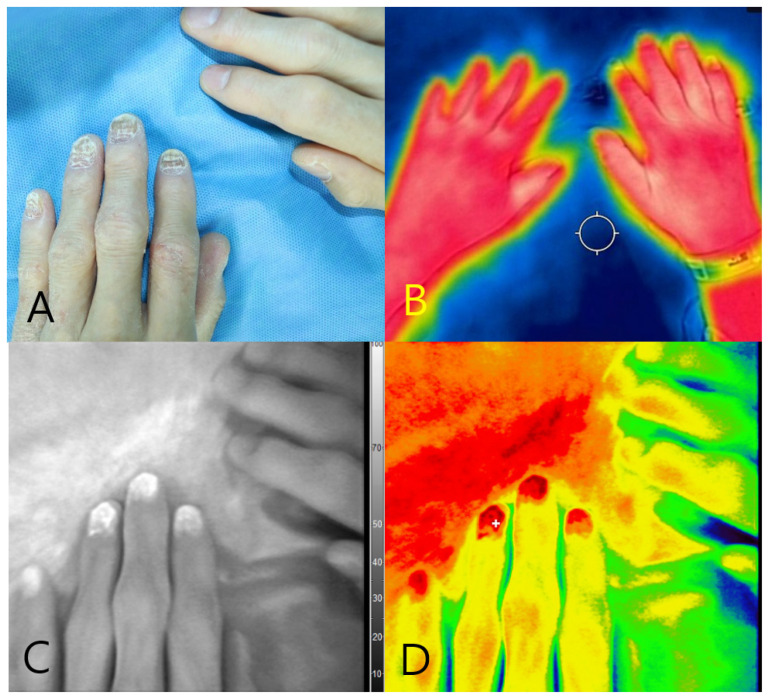
Multimodal imaging of onychomycosis in a 59-year-old female patient. (**A**) Clinical photograph showing dystrophic changes of the second to fourth fingernails on the left hand. (**B**) Thermal image demonstrating overall symmetrical temperature distribution, with focal hyperthermia in the affected nails. (**C**) Near-infrared fluorescence image showing autofluorescence in the second to fourth fingernails of the left hand. (**D**) Pseudocolor overlay confirming fluorescence signals at the culture-positive nails, correlating with *Trichophyton rubrum* infection.

**Figure 6 diagnostics-15-02221-f006:**
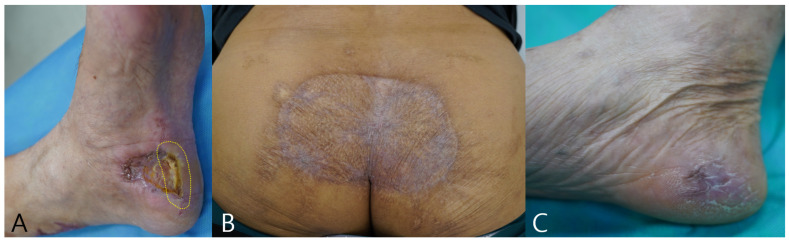
Long-term outcomes. (**A**) Case 1 at 3 months: residual slough at the prior fluorescence-positive margin (yellow dashed ellipse), subsequently debrided and healed. (**B**) Case 2 at 12 months: durable epithelialization. (**C**) Case 3 at 12 months: durable epithelialization.

**Table 1 diagnostics-15-02221-t001:** Distribution of wound types among study participants.

Wound Type	Number of Patients (%)
Diabetic foot ulcers	13 (39.4%)
Necrotic wounds	8 (24.2%)
Cancer-related wounds	6 (18.2%)
Burn wounds	6 (18.2%)
Total	33 (100%)

**Table 2 diagnostics-15-02221-t002:** Contingency table comparing fluorescence imaging results with wound culture outcomes.

Culture Result	Fluorescence Positive	Fluorescence Negative	Total
Positive (Infected)	18 (TP)	5 (FN)	23
Negative (No infection)	2 (FP)	8 (TN)	10
Total	20	13	33

Note: Values represent the number of wounds per category. TP, true positive; FN, false negative; FP, false positive; TN, true negative.

**Table 3 diagnostics-15-02221-t003:** Stratification of fluorescence signal intensity by microbial burden and clinical severity.

Signal Intensity	Mean Microbial Burden (CFU/mL)	CFU Range	Clinical Severity	Number of Cases
High	9.2 × 10^6^	8.5 × 10^6^–1.1 × 10^7^	Severe	6
Moderate	5.8 × 10^5^	4.7 × 10^5^–6.3 × 10^5^	Moderate	10
Low	2.9 × 10^4^	2.0 × 10^4^–3.5 × 10^4^	Mild	2

CFU, colony-forming unit.

**Table 4 diagnostics-15-02221-t004:** Correlation between fluorescence signal intensity and clinical indicators.

Clinical Indicator	Correlation Coefficient (r)	*p*-Value
Exudate amount	0.72	0.004
Swelling	0.68	0.006
Foul odor	0.68	0.008
Induration	0.65	0.081
Pain	0.59	0.137

**Table 5 diagnostics-15-02221-t005:** Wound Surface Temperature According to Fluorescence Signal.

Fluorescence Signal	Mean Temperature (°C)	Standard Deviation	Observed Thermal Pattern
Positive	36.8	±0.3	Localized hyperthermia or near fluorescent zones
Negative	36.5	±0.2	No distinct thermal elevation

**Table 6 diagnostics-15-02221-t006:** Fluorescence imaging detection by pathogen species.

Pathogen (Culture Positive)	Cases (*n*)	Detected by Fluorescence (*n*)	Detection Rate
*Pseudomonas aeruginosa*	6	6	100%
*Staphylococcus aureus*	8	6	75.00%
*Enterococcus faecalis*	1	0	0%
*Klebsiella pneumoniae*	2	1	50.00%
*Escherichia coli*	2	1	50.00%
Fungal isolates (e.g., *Candida* spp.)	4	4	100%

Note: The table shows the number of wounds infected with each organism (per culture result) and the number of wounds correctly identified (fluorescence-positive) using the Fluobeam device. The detection rate was calculated as the percentage of culture-positive cases with positive fluorescent signals.

**Table 7 diagnostics-15-02221-t007:** Species-level detectability with literature-based fluorescent metabolites.

Pathogen	Classification	Detectability	Known Fluorescent Metabolites	References
*Pseudomonas aeruginosa*	Gram-negative aerobic rod	High (100%)	Pyoverdine, Pyocyanin	[17,27]
*Staphylococcus aureus*	Gram-positive facultative anaerobic coccus	Moderate (75%)	Porphyrins (coproporphyrin, heme biosynthetic intermediates)	[2,27]
*Enterococcus faecalis*	Gram-positive facultative anaerobic coccus	Low (0%)	None identified (minimal fluorophore production)	[18]
*Klebsiella pneumoniae*	Gram-negative facultative anaerobic rod	Moderate (50%)	Porphyrins (protoporphyrin IX, coproporphyrin I)	[29]
*Escherichia coli*	Gram-negative facultative anaerobic rod	Moderate (50%)	Porphyrins (protoporphyrin IX)	[29]
*Candida* spp. (yeasts)	Fungus (yeast)	High (100%)	Unidentified (potentially porphyrin-related fluorophores)	[2,28]

**Table 8 diagnostics-15-02221-t008:** Concise comparison of adjunctive tools for wound infection assessment at the point of care and in the laboratory.

Tool/ Modality	What It Measures	Real-Time Spatial Map	Species Identification	Typical Turnaround	Key Advantages/Limitations	References
Fluobeam^®^ + FLIR^®^ (dual modality)	Near-infrared microbial autofluorescence (780 nm excitation/820 nm detection) and skin surface temperature (infrared)	Yes	No	Seconds	Advantages: Real-time bedside localization of microbial foci; adjunct mapping of perilesional spread. Limitations: Off-label for microbial detection; organism-dependent detectability; no spectral identification; small ΔT (~0.3 °C) without device-level thermal calibration; sensitive to ambient conditions and vascular status.	[2,32,37]
MolecuLight^®^	Porphyrins (red) and pyoverdine (cyan) under 405 nm excitation	Yes	Limited	Seconds	Advantages: Portable POC workflow; robust clinical evidence for bacterial-burden mapping; guides debridement and targeted culture. Limitations: Detectability varies by organism (non-porphyrin/low-metabolic species less visible); ambient-light constraints; no species-level ID.	[12,14]
PCR	Microbial nucleic acids	No	Yes	Hours	Advantages: High analytic sensitivity/specificity; species/target detection; potential resistance marker genotyping. Limitations: Cost/infrastructure needs; contamination risk; does not confirm viability; no spatial mapping; longer turnaround.	[4]
Culture	Growth of viable organisms from tissue/swab	No	Yes	Days	Advantages: Organism identification with antimicrobial susceptibility; clinical reference standard. Limitations: Slow (days); false-negatives with fastidious/biofilm organisms; sampling error; no spatial map.	[4]

FLIR^®^, forward-looking infrared; POC, point-of-care; ID, species identification; NIR, near-infrared; ΔT, temperature difference; PCR, polymerase chain reaction.

**Table 9 diagnostics-15-02221-t009:** Bacterial Species with Known Autofluorescent Properties.

Category	Species
Gram-negative aerobic species	*Pseudomonas aeruginosa* *Escherichia coli* *Proteus mirabilis* *Proteus vulgaris* *Enterobacter cloacae* *Serratia marcescens* *Acinetobacter baumannii* *Klebsiella pneumoniae* *Klebsiella oxytoca* *Morganella morganii* *Stenotrophomonas maltophilia* *Citrobacter koseri* *Citrobacter freundii* *Aeromonas hydrophilia* *Alcaligenes faecalis* *Pseudomonas putida*
Gram-positive aerobic species	*Staphylococcus aureus* *Staphylococcus epidermidis* *Staphylococcus lugdunensis* *Staphylococcus capitis* *Corynebacterium striatum* *Bacillus cereus* *Listeria monocytogenes*
Anaerobic species	*Bacteroides fragilis* *Clostridium perfringens* *Peptostreptococcus anaerobius* *Propionibacterium acnes* *Veillonella parvula*

## Data Availability

The data presented in this study are available on request from the corresponding author due to restrictions related to patient privacy and institutional ethical guidelines.

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
