# Peer review of "New Application for the Early Detection of Wound Infections Using a Near-Infrared Fluorescence Device and Forward-Looking Thermal Camera"

_diagnostics, 2025, doi:10.3390/diagnostics15172221_

Round 1

Reviewer 1 Report

Comments and Suggestions for Authors
  • This research presents an encouraging dual-modality method for identifying wound infections, highlighting its advantages in real-time, noninvasive imaging and sensitivity to specific pathogens. Nonetheless, the limitations (such as sample size, organism-dependent detection, and minimal thermal variations) need to be explicitly acknowledged. Some of the key limitations are listed below.
  1. The small sample size (n=33) and retrospective design limit the findings' generalizability; a prospective study with a larger, diverse cohort (>100) is recommended. Justification for exclusion criteria like "excessive exudate" is needed, as it may bias results. The authors should acknowledge these limitations in the Discussion and propose a future study to validate results across various wound types (e.g., burns, diabetic ulcers).
  2. While the sensitivity (78.3%) and specificity (80.0%) are promising, they fall short of >90% reported in prior studies. The Discussion should explore reasons for this, address the low negative predictive value (61.5%), and recommend confirmatory tests (e.g., PCR) for fluorescence-negative wounds, especially concerning pathogens like Enterococcus faecalis.
  3. Detection of Pseudomonas aeruginosa and Candida spp. is effective, but sensitivity for gram-positive and anaerobic species is poor. Clearer differentiation of microbial properties is needed, with a new section on pathogen-specific detection limits in the Results/Discussion. Tables should summarize known autofluorescent pathogens, citing relevant literature to support species-specific fluorescence claims.
  4. Data on the off-label use of the Fluobeam device concerning bacterial detection calibration is essential. The minimal thermal difference (0.3°C) raises questions about its clinical relevance, especially in vascular patients. A Methods paragraph detailing device calibration and clinical implications of thermal differences should be included.
  5. Key terms like “autofluorescence” and “microbial burden” must be clearly defined and consistently used throughout the manuscript.
  6. A comparative table of Fluobeam/FLIR versus existing point-of-care tools (e.g., PCR, MolecuLight) should be added, addressing their advantages and limitations along with relevant studies for context.
  7. Lastly, ensure all claims about pathogen fluorescence are grounded in primary literature and update the introduction with recent advances in fluorescence imaging. Include up-to-date reviews and primary sources for specific claims.
  8.  

Author Response

Response to Reviewer 1

Comment 1

The small sample size (n=33) and retrospective design limit the findings' generalizability; a prospective study with a larger, diverse cohort (>100) is recommended. Justification for exclusion criteria like "excessive exudate" is needed, as it may bias results. The authors should acknowledge these limitations in the Discussion and propose a future study to validate results across various wound types (e.g., burns, diabetic ulcers).

Response

We appreciate this comment and agree. We have revised the manuscript to (i) explicitly acknowledge the preliminary, single-center, retrospective design and small sample size; (ii) justify the exclusion of wounds with excessive exudate on methodological grounds; and (iii) outline a plan for a prospective, adequately powered, multicenter validation across diverse wound types.

Revisions in the manuscript (Highlighted in red)

  1. Materials and Methods — Section 2.3
    Wounds with excessive exudate were excluded to avoid optical attenuation and to ensure reproducible region-of-interest quantification under the standardized protocol.
  2. Discussion
    This study should be interpreted as a preliminary, single-center, retrospective analysis with a small sample size (n=33), which limits precision and external validity.
    Because the cohort was enriched for clinically suspected infections and we excluded wounds with excessive exudate, the reported performance may be overestimated relative to unselected wound populations.
  3. Discussion — Limitations
    This study had several limitations. First, it was a preliminary, single-center, retrospective study with a small sample size (n=33), which limits precision and generalizability across heterogeneous wound types and microbial spectra. Inclusion was restricted to clinically suspected infections; subclinical infections and non-infectious inflammatory conditions were not systematically included, which may bias case mix and overestimate diagnostic performance relative to unselected populations. Second, to avoid optical attenuation and ensure reproducible ROI placement, we excluded wounds with excessive exudate; together with enrichment for clinically apparent infection, this may introduce selection bias. Third, although fluorescence intensity correlated with infection severity, the device lacks species-level resolution; species-level detection proportions were underpowered for several taxa (e.g., rare isolates) and are reported descriptively without inferential claims. Fourth, thermal findings were based on modest surface temperature differences (~0.3 °C) acquired without blackbody-traceable device calibration; such small deltas may fall within physiologic or measurement variability and can be influenced by environmental or vascular factors (e.g., peripheral vascular disease). We did not stratify performance by wound location (hair-bearing vs. glabrous skin), skin phototype, or operator experience, which may affect signal visibility and ROI placement, and we did not perform standalone-versus-combined ROC analyses, so the incremental benefit of the dual modality was not quantified. In addition, Fluobeam® was used off-label for bacterial/fungal autofluorescence without microorganism-specific calibration; thermal–fluorescence co-localization was assessed qualitatively without pre-specified spatial metrics; and correlations between fluorescence and clinical signs were unadjusted and susceptible to confounding (e.g., wound size, chronicity, comorbidities). Finally, culture-based reference standards have limitations, particularly for fastidious or biofilm-associated organisms, and routine anaerobic culture was not performed in all cases. To mitigate these limitations, we plan a prospective, adequately powered, multicenter study (target >100 participants) spanning diverse wound etiologies (burns, diabetic foot ulcers, malignancy-associated wounds), with consecutive enrollment (including subclinical infections and non-infectious mimics), pre-specified endpoints and subgroup powering, blinded image interpretation, expanded microbiology including anaerobes and molecular assays, and device calibration and repeatability assessments for fluorescence and thermal imaging.

Comment 2

While the sensitivity (78.3%) and specificity (80.0%) are promising, they fall short of >90% reported in prior studies. The Discussion should explore reasons for this, address the low negative predictive value (61.5%), and recommend confirmatory tests (e.g., PCR) for fluorescence-negative wounds, especially concerning pathogens like Enterococcus faecalis.

Response

We agree. In the Discussion, we now explain why our sensitivity/specificity are lower than some prior reports, explicitly address the modest NPV (61.5%), and provide a clinical recommendation for confirmatory testing in fluorescence-negative wounds—particularly when non-fluorogenic pathogens (e.g., Enterococcus spp. or anaerobes) are plausible. No new cases were added (final n=33).

Revisions in the manuscript (Highlighted in red)

  1. Discussion

Rather than functioning solely as a binary diagnostic tool, Fluobeam® demonstrated the capacity to provide semi-quantitative information regarding microbial burden. In our study, the fluorescence signal intensity closely reflected the bacterial density within the wound, supporting its use as a dynamic marker of infection severity [2,3,21]. As shown in Table 2, the device achieved a sensitivity of 78.3% (18/23) and a specificity of 80.0% (8/10) for detecting culture-positive infections, suggesting a robust diagnostic performance with an acceptable false-positive rate [5,30]. However, our sensitivity (78.3%) and specificity (80.0%) are below the >90% reported in some prior studies. Several factors may account for this difference: the preliminary, small single-center cohort; a case mix dominated by chronic, previously debrided wounds; the off-label use of a device without species-specific bacterial-detection calibration; reliance on surface signals that can miss deep-seated or biofilm-associated infections; and lack of routine anaerobic culture in all cases. Consistent with these factors, the negative predictive value was modest (61.5%), indicating that negative fluorescence does not rule out infection. Accordingly, for fluorescence-negative wounds with persistent clinical suspicion—particularly when non-fluorogenic pathogens (e.g., Enterococcus spp. or anaerobes) are plausible—we recommend confirmatory testing (tissue culture including anaerobic coverage and, when available, molecular assays such as PCR).

Comment 3

Detection of Pseudomonas aeruginosa and Candida spp. is effective, but sensitivity for gram-positive and anaerobic species is poor. Clearer differentiation of microbial properties is needed, with a new section on pathogen-specific detection limits in the Results/Discussion. Tables should summarize known autofluorescent pathogens, citing relevant literature to support species-specific fluorescence claims.

Response

We appreciate this comment and agree. We revised the manuscript to: (i) add a dedicated Results subsection (Section 3.7, Species-Specific Detection Limits and Autofluorescent Properties) and Table 7 summarizing species-level detection under our 780-nm excitation/820-nm detection protocol, with descriptive detectability tiers (High ≥90%, Moderate 50–89%, Low <50%) and an explicit note that gram-positive cocci show lower detection and anaerobes were not systematically assessed; (ii) expand the Discussion to provide mechanistic context (e.g., pyoverdine in Pseudomonas aeruginosa, porphyrins in Staphylococcus aureus), to caution that most reported bacterial fluorophore peaks lie in the violet/visible range and that near-infrared excitation/detection without spectral decomposition cannot support species identification, and to add clinical guidance (confirmatory tissue sampling with anaerobic culture and, where available, PCR when fluorescence is negative; use fluorescence-positive zones to target debridement and culture sampling); and (iii) align citations to primary sources and include a table note clarifying that the tiers are descriptive and cohort-specific, not analytical limits of detection.

Revisions in the manuscript (Highlighted in red)

  1. Result

3.7. Species-Specific Detection Limits and Autofluorescent Properties

Table 7 summarizes the pathogens isolated in this cohort, their detection rates within the study, and autofluorescent metabolites and properties reported in the literature under the 780 nm excitation and 820 nm emission detection settings used. Detection rates were categorized as High (≥ 90%), Moderate (50–89%), and Low (< 50%), based on the proportion of fluorescence-positive results among culture-confirmed wounds for each species. These detection rates are descriptive and cohort-specific, and do not represent analytical limits of detection or comprehensive device performance metrics. Organisms with well-characterized endogenous fluorophores—such as P. aeruginosa and S. aureus—demonstrated higher detection rates. In contrast, Candida spp. showed high detectability in this cohort, but their endogenous fluorophores remain poorly characterized; current evidence suggests the presence of putative porphyrin-related fluorophores, though this requires further investigation. Gram-positive cocci exhibited lower detection rates. Anaerobic species were not systematically evaluated and are not represented.

Pathogen

Classification

Detectability

Known Fluorescent Metabolites

References

Pseudomonas aeruginosa

Gram-negative aerobic rod

High

(100%)

Pyoverdine, Pyocyanin

[17,27]

Staphylococcus aureus

Gram-positive facultative anaerobic coccus

Moderate (75%)

Porphyrins
(coproporphyrin, heme biosynthetic intermediates)

[2,27]

Enterococcus faecalis

Gram-positive facultative anaerobic coccus

Low

(0%)

None identified
(minimal fluorophore production)

[18]

Klebsiella pneumoniae

Gram-negative facultative anaerobic rod

Moderate (50%)

Porphyrins
(protoporphyrin IX, coproporphyrin I)

[29]

Escherichia coli

Gram-negative facultative anaerobic rod

Moderate (50%)

Porphyrins (protoporphyrin IX)

[29]

Candida spp. (yeasts)

Fungus (yeast)

High

(100%)

Unidentified (potentially porphyrin-related fluorophores)

[2,28]

Table 7. Species-level detectability with literature-based fluorescent metabolites

  1. Discussion

These species-level patterns are consistent with fluorescence biology reported for wound pathogens: organisms that biosynthesize strong endogenous fluorophores—such as pyoverdine in P. aeruginosa and porphyrins in S. aureus—tend to be more readily detected, whereas taxa with weak or absent fluorophore production (e.g., gram-positive cocci such as Enterococcus spp. and many anaerobes) show poorer detectability under our 780/820-nm acquisition protocol. Because most reported bacterial fluorophores peak in the violet/visible range, extrapolation to near-infrared excitation should be cautious; the device does not provide spectral decomposition, and species identification is not feasible from fluorescence alone [14–18,28]. Accordingly, Table 7 is provided as an interpretive aid that links our observed detection proportions with literature-based autofluorescent metabolites and typical visible-range characteristics rather than as a species-identification key.

From a clinical standpoint, a negative fluorescence finding should not be taken to exclude infection when low- or non-fluorogenic organisms are plausible (e.g., Enterococcus spp. or anaerobes); confirmatory testing with tissue sampling—including anaerobic culture—and, where available, molecular assays (e.g., PCR) is recommended. Conversely, fluorescence-positive areas can be used to target debridement and to localize culture sampling to the highest-yield zone, potentially improving microbiological confirmation and procedural efficiency [2,13,14,27].

Comment 4

Data on the off-label use of the Fluobeam device concerning bacterial detection calibration is essential. The minimal thermal difference (0.3°C) raises questions about its clinical relevance, especially in vascular patients. A Methods paragraph detailing device calibration and clinical implications of thermal differences should be included.

Response

We appreciate this important comment and agree. We revised the manuscript to (i) explicitly document the off-label use of Fluobeam® for bacterial/fungal autofluorescence and our acquisition standardization; (ii) detail how thermal images were acquired and interpreted in the absence of blackbody-traceable calibration; and (iii) clarify the clinical implications and limitations of the small thermal contrast (~0.3 °C), especially in vascular patients. Specifically, the Methods now describe fixed distance/exposure/perpendicular orientation under controlled ambient conditions, ROI placement and a relative fluorescence index, and thermal settings (emissivity 0.98, co-registered ROIs, ROI-mean temperatures). We state that no manufacturer calibration for microbial detection and no blackbody-traceable thermal calibration were available/performed in this retrospective study; therefore small absolute temperature differences (~0.3 °C) were treated as indeterminate and we prioritized pattern-based interpretation (co-localization with fluorescence, perilesional extension, serial change). In the Results, we explicitly note that the ~0.3 °C mean difference is modest and should be interpreted cautiously. The Discussion adds a paragraph on the clinical implications of small thermal deltas and outlines prospective calibration and repeatability studies. The Limitations were updated to acknowledge off-label use without microorganism-specific calibration and the lack of blackbody-traceable thermal calibration.

Revisions in the manuscript (Highlighted in red)

2.2. Imaging System

Use of Fluobeam® for bacterial/fungal autofluorescence in this study was off-label; no manufacturer-provided calibration for microbial detection was available. To standardize acquisition, imaging was performed in a controlled operating room (21–24 °C; 40–60% relative humidity) under consistent ambient lighting, using a fixed working distance (~20 cm), fixed exposure settings and detector gain, and a perpendicular camera orientation to the wound surface.

2.4. Imaging Procedure

All imaging procedures were conducted in a controlled operating room maintained at a stable ambient temperature (21–24 °C) and relative humidity (40–60%) under consistent ambient lighting. … All images were acquired at a fixed distance of approximately 20 cm from the wound surface with fixed exposure settings and detector gain and a perpendicular camera orientation to the wound surface.
… A second round of fluorescence imaging was performed immediately using identical acquisition parameters.
Autofluorescent regions were identified … quantified using the manufacturer’s proprietary software; circular regions of interest (ROIs; ~1 cm²) were placed on peak signal within the wound and on adjacent clinically uninvolved skin, and a relative fluorescence index (wound/reference ROI ratio) was computed to mitigate inter-frame variability.
Thermal imaging was performed immediately after the fluorescence imaging under identical environmental and positional conditions. Skin emissivity was set to 0.98; ROIs (~1 cm²) were co-registered to the fluorescence maps, and temperatures are reported as ROI means. No blackbody-traceable thermal calibration was performed in this retrospective study; accordingly, small absolute temperature differences on the order of a few tenths of a degree Celsius (≈0.3 °C) were treated as indeterminate, and interpretation prioritized spatial patterns (co-localization with fluorescence, perilesional extension, and—where available—serial change) over single absolute thresholds.

3.5. Thermal Imaging Findings

However, the mean absolute temperature difference was modest (~0.3 °C) and, in the absence of device-level calibration, should be interpreted cautiously; emphasis was placed on spatial concordance and perilesional spread rather than the magnitude of the delta.

  1. Discussion

Clinical implications of small thermal differences. The average thermal contrast observed in this cohort was small (~0.3 °C). In patients with vascular compromise (e.g., peripheral arterial disease) or under cool ambient conditions, baseline skin temperature and thermal gradients may be blunted, further limiting the diagnostic value of small absolute deltas. Therefore, in this study thermal imaging primarily served to map spatial patterns (co-localization with fluorescence and perilesional extension) rather than to apply absolute thresholds. Future work will incorporate blackbody-traceable calibration and repeated-measures assessments to define device-specific reproducibility and clinically meaningful temperature differentials [33–35].

  1. Discussion - Limitations

Fourth, thermal findings were based on modest surface temperature differences (~0.3 °C) acquired without blackbody-traceable device calibration; such small deltas may fall within physiologic or measurement variability and can be influenced by environmental or vascular factors (e.g., peripheral vascular disease). We did not stratify performance by wound location (hair-bearing vs. glabrous skin), skin phototype, or operator experience, which may affect signal visibility and ROI placement, and we did not perform standalone-versus-combined ROC analyses, so the incremental benefit of the dual modality was not quantified. In addition, Fluobeam® was used off-label for bacterial/fungal autofluorescence without microorganism-specific calibration; thermal–fluorescence co-localization was assessed qualitatively without pre-specified spatial metrics; and correlations between fluorescence and clinical signs were unadjusted and susceptible to confounding (e.g., wound size, chronicity, comorbidities). Finally, culture-based reference standards have limitations, particularly for fastidious or biofilm-associated organisms, and routine anaerobic culture was not performed in all cases. To mitigate these limitations, we plan a prospective, adequately powered, multicenter study (target >100 participants) spanning diverse wound etiologies (burns, diabetic foot ulcers, malignancy-associated wounds), with consecutive enrollment (including subclinical infections and non-infectious mimics), pre-specified endpoints and subgroup powering, blinded image interpretation, expanded microbiology including anaerobes and molecular assays, and device calibration and repeatability assessments for fluorescence and thermal imaging.

Comment 5

Key terms like “autofluorescence” and “microbial burden” must be clearly defined and consistently used throughout the manuscript.

Response

We agree. To ensure clarity and consistency, we (i) added a concise terminology note that defines fluorescence, autofluorescence, and microbial burden; and (ii) replaced all instances of “bacterial load,” “microbial load,” and “bacterial burden” with microbial burden across the manuscript (text, section headings, tables/figure legends). All new or revised text is highlighted in the manuscript.

Revisions in the manuscript (Highlighted in red)

  1. Introduction

In this manuscript, fluorescence refers to the optical phenomenon; autofluorescence denotes endogenous fluorescence emitted by microbial or tissue fluorophores without exogenous contrast agents, and microbial burden refers to the quantitative density of viable microorganisms measured from tissue culture and expressed as CFU/mL.

Comment 6

A comparative table of Fluobeam/FLIR versus existing point-of-care tools (e.g., PCR, MolecuLight) should be added, addressing their advantages and limitations along with relevant studies for context.

Response

Thank you for this helpful suggestion. We agree that contextualizing our dual-modality approach against established point-of-care tools will aid readers. We have therefore (i) added a concise comparison table that contrasts Fluobeam® + FLIR®, MolecuLight®, PCR, and culture with respect to what each measures, spatial mapping capability, species identification, turnaround time, key advantages/limitations, and representative references; and (ii) inserted a brief Discussion paragraph that explains how these modalities complement one another in clinical workflows.

Revisions in the manuscript (Highlighted in red)

To place our dual-modality approach in context, we added a concise comparison of adjunct tools (Table 8). Fluobeam® + FLIR® provides rapid bedside mapping of microbial foci and perilesional spread but is off-label for bacterial detection, offers organism-dependent detectability, lacks spectral identification, and small absolute thermal deltas (~0.3 °C) warrant cautious interpretation without device-level calibration [2,33–35]. MolecuLight® is a purpose-built, portable point-of-care fluorescence platform with robust clinical evidence for bacterial-burden mapping and guidance of debridement/culture, although visibility depends on porphyrin/cyan fluorophore production and ambient-light constraints; species-level identification is not provided [12,14]. PCR delivers high analytic sensitivity/specificity and can detect targets/resistance but lacks viability confirmation, spatial mapping, and rapid turnaround compared with imaging [4]. Culture remains the clinical reference for organism identification and antimicrobial susceptibility, albeit with slower turnaround and potential false-negatives in fastidious/biofilm organisms [4]. Taken together, Table 8 highlights complementary roles: imaging rapidly maps where to sample and treat, whereas laboratory tests determine what is present and how to target therapy.

Table 8. Concise comparison of adjunctive tools for wound infection assessment at the point of care and in the laboratory

Tool /
Modality

What it measures

Real-time spatial map

Species identification

Typical turnaround

Key advantages/limitations

References

Fluobeam® + FLIR®
(dual modality)

Near-infrared microbial autofluorescence (780 nm excitation / 820 nm detection) and skin surface temperature (infrared)

Yes

No

Seconds

Advantages: Real-time bedside localization of microbial foci; adjunct mapping of perilesional spread.
Limitations: Off-label for microbial detection; organism-dependent detectability; no spectral identification; small ΔT (~0.3 °C) without device-level thermal calibration; sensitive to ambient conditions and vascular status.

[2,33,34]

MolecuLight®

Porphyrins (red) and pyoverdine (cyan) under 405 nm excitation

Yes

Limited

Seconds

Advantages: Portable POC workflow; robust clinical evidence for bacterial-burden mapping; guides debridement and targeted culture.
Limitations: Detectability varies by organism (non-porphyrin/low-metabolic species less visible); ambient-light constraints; no species-level ID.

[12,14]

PCR

Microbial nucleic acids

No

Yes

Hours

Advantages: High analytic sensitivity/specificity; species/target detection; potential resistance marker genotyping.
Limitations: Cost/infrastructure needs; contamination risk; does not confirm viability; no spatial mapping; longer turnaround.

[4]

Culture

Growth of viable organisms from tissue/swab

No

Yes

Days

Advantages: Organism identification with antimicrobial susceptibility; clinical reference standard.
Limitations: Slow (days); false-negatives with fastidious/biofilm organisms; sampling error; no spatial map.

[4]

FLIR®, forward-looking infrared; POC, point-of-care; ID, species identification; NIR, near-infrared; ΔT, temperature difference; PCR, polymerase chain reaction.

Comment 7

Lastly, ensure all claims about pathogen fluorescence are grounded in primary literature and update the introduction with recent advances in fluorescence imaging. Include up-to-date reviews and primary sources for specific claims.

Response

We have updated the Introduction with a new paragraph summarizing recent advances in clinical fluorescence imaging (point-of-care violet/visible excitation platforms, label-free spectroscopic methods, and fluorescence-lifetime imaging) and briefly linking characteristic emissions to underlying fluorophores (pyoverdine in Pseudomonas aeruginosa, porphyrins in Staphylococcus aureus). We cite up-to-date reviews and primary sources. In addition, species-specific statements elsewhere in the manuscript (e.g., Table 7 and Discussion) are aligned to primary or foundational literature.

Revisions in the manuscript (Highlighted in red)

Recent advances in clinical wound imaging include point-of-care violet and visible excitation platforms that visualize porphyrin-related red emission and cyan emission associated with Pseudomonas aeruginosa in real time, label-free spectroscopic methods, and fluorescence-lifetime imaging capable of fingerprinting microbial metabolic state [2,7–9,12–14]. At the molecular level, Pseudomonas aeruginosa produces fluorescent siderophores such as pyoverdine, whereas Staphylococcus aureus accumulates porphyrins, which together explain these characteristic emission colors [15–18]. Because most reported bacterial fluorophores peak in the violet or visible range, extension to near-infrared excitation and detection as used in this study should be interpreted with caution and does not enable species identification without spectral decomposition.

Reviewer 2 Report

Comments and Suggestions for Authors

This study explores a novel combination of near-infrared fluorescence imaging (Fluobeam®) and thermal imaging (FLIR®) for detecting wound infections, which holds clinical promise. However, several critical issues undermine its robustness and generalizability, as detailed below:

1. Limited Sample Size and Selection Bias

The study includes only 33 patients, with diabetic foot ulcers (39.4%) comprising the largest subgroup, followed by necrotic wounds (24.2%), cancer-related wounds (18.2%), and burns (18.2%) . This small and unevenly distributed sample fails to support claims about "chronic wounds" broadly. Rare pathogens (e.g., Enterococcus faecalis, n=1) yield statistically underpowered detection rates, making conclusions about organism-specific sensitivity unreliable .

Moreover, the retrospective design—limited to "clinically suspected infections"—introduces selection bias. By excluding subclinical infections or non-infectious inflammatory wounds, the study may overestimate diagnostic performance, as it does not test the tools’ ability to differentiate true infections from mimics.

2. Inadequate Methodological Details

Imaging protocols lack rigor: While Fluobeam®’s excitation (780 nm) and detection (820 nm) wavelengths are noted, the proprietary software used to quantify fluorescence intensity is not described . Without transparency on algorithms or calibration, results cannot be validated or replicated. For FLIR® thermal imaging, the 0.3°C temperature difference between fluorescence-positive and negative wounds (36.8°C vs. 36.5°C) is presented as meaningful, but no data on measurement reproducibility are provided to support this .

Microbiological methods are incomplete: Cultures focus on aerobic and facultative anaerobic pathogens but exclude anaerobes—common in chronic wounds—risking false negatives . Additionally, the criteria for classifying "semi-quantitative bacterial loads" (e.g., defining "high" vs. "low" CFU/mL) are absent, weakening correlations drawn between fluorescence intensity and infection severity .

3. Superficial Analysis of Results

Pathogen-specific variability is poorly explained: The study reports stark differences in detection rates but provides little mechanistic insight . It attributes this to "metabolic fluorescence properties" but does not link specific pathogens to their fluorescent metabolites  or test whether wavelength adjustments could improve detection .

Correlations ignore confounders: Fluorescence intensity correlates with exudate (r=0.72), swelling (r=0.68), and odor (r=0.68), but these associations are not adjusted for confounding variables like wound size, duration, or patient comorbidities . This limits conclusions about whether fluorescence truly reflects infection rather than unrelated wound characteristics.

Thermal-fluorescence spatial relationships are understudied: The claim that thermal signals "extend beyond fluorescent zones" is based on qualitative observations, not quantitative analyses (e.g., frequency of this pattern or its association with subsequent infection spread) .

4. Overstated Conclusions

The authors conclude that combining Fluobeam® and FLIR® "improves infection detection," but no direct comparison of standalone vs. combined performance (e.g., ROC curves) is provided . Similarly, claims about "enhancing intraoperative decision-making" rely on 4 case studies, which are insufficient to demonstrate clinical utility .

Limitations are underplayed: The study does not address how wound location (e.g., hairy vs. glabrous skin) or operator experience might affect results. It also fails to contextualize findings against existing tools (e.g., MolecuLight i:X™), leaving readers unsure of Fluobeam®’s unique value .

Author Response

Response to Reviewer 2

Comment 1

  1. Limited Sample Size and Selection Bias

The study includes only 33 patients, with diabetic foot ulcers (39.4%) comprising the largest subgroup, followed by necrotic wounds (24.2%), cancer-related wounds (18.2%), and burns (18.2%) . This small and unevenly distributed sample fails to support claims about "chronic wounds" broadly. Rare pathogens (e.g., Enterococcus faecalis, n=1) yield statistically underpowered detection rates, making conclusions about organism-specific sensitivity unreliable .

 Moreover, the retrospective design—limited to "clinically suspected infections"—introduces selection bias. By excluding subclinical infections or non-infectious inflammatory wounds, the study may overestimate diagnostic performance, as it does not test the tools’ ability to differentiate true infections from mimics.

Response

Thank you for this important critique. We agree that the small, unevenly distributed sample and retrospective inclusion of clinically suspected infections limit generalizability and may introduce selection bias. In response, we (i) tempered claims about “chronic wounds” to the population actually studied (chronic wounds with clinically suspected infection); (ii) explicitly state in the Results and Discussion that species-level detection estimates are underpowered (e.g., Enterococcus faecalis n=1) and should be interpreted descriptively; and (iii) expanded the Limitations and Future Work to outline a prospective, adequately powered, multicenter study with consecutive enrollment across wound etiologies, inclusion of subclinical infections and non-infectious inflammatory mimics, blinded image interpretation, and expanded anaerobic microbiology. These revisions clarify scope, acknowledge potential overestimation of performance due to case enrichment, and provide a concrete validation plan.

Revisions in the manuscript

1) Introduction

…This highlights the need for more accurate and real-time diagnostic tools to facilitate the early detection and timely management of wound infections [4,5]. Accordingly, the present work focuses on chronic wounds with clinically suspected infection managed in a surgical setting, and findings should be interpreted within this clinical context.

2) Results – 3.1. Patient Demographics

…in accordance with the institutional protocol. Given the small and uneven subgroup sizes across wound etiologies, no inferential comparisons by wound type were performed.

3) Results – 3.6. Microbiological Findings

Because several taxa were represented by very small numbers (e.g., Enterococcus faecalis n=1), species-level detection proportions are reported descriptively without inferential claims.

4) Discussion – Limitations

This study had several limitations. First, it was a preliminary, single-center, retrospective study with a small sample size (n=33), which limits precision and generalizability across heterogeneous wound types and microbial spectra. Inclusion was restricted to clinically suspected infections; subclinical infections and non-infectious inflammatory conditions were not systematically included, which may bias case mix and overestimate diagnostic performance relative to unselected populations.

5) Discussion – Future validation

…device calibration and repeatability assessments for fluorescence and thermal imaging. The planned study will use consecutive enrollment across wound etiologies (including subclinical infections and non-infectious mimics), pre-specified powering for subgroup analyses, blinded image interpretation, and expanded anaerobic culture/molecular testing to quantify both sensitivity and specificity in an unselected population.

6) Conclusions

This study demonstrates a novel use of the Fluobeam® near-infrared fluorescence system for detecting bacterial and fungal infections in chronic wounds with clinically suspected infection through autofluorescence imaging. …

Comment 2

  1. Inadequate Methodological Details

 Imaging protocols lack rigor: While Fluobeam®’s excitation (780 nm) and detection (820 nm) wavelengths are noted, the proprietary software used to quantify fluorescence intensity is not described . Without transparency on algorithms or calibration, results cannot be validated or replicated. For FLIR® thermal imaging, the 0.3°C temperature difference between fluorescence-positive and negative wounds (36.8°C vs. 36.5°C) is presented as meaningful, but no data on measurement reproducibility are provided to support this.

 Microbiological methods are incomplete: Cultures focus on aerobic and facultative anaerobic pathogens but exclude anaerobes—common in chronic wounds—risking false negatives . Additionally, the criteria for classifying "semi-quantitative bacterial loads" (e.g., defining "high" vs. "low" CFU/mL) are absent, weakening correlations drawn between fluorescence intensity and infection severity.

Response

Thank you for these important points. We have revised the Methods to improve transparency and reproducibility. Specifically, we now (i) describe how fluorescence intensity was quantified with vendor software using ROI-based measurements and normalization (relative fluorescence index, RFI), while noting the proprietary nature of the processing algorithms; (ii) detail fixed acquisition parameters, ROI co-registration across sessions, and multiple-frame capture; (iii) clarify FLIR® settings (emissivity 0.98), ROI co-registration, and that no blackbody-traceable thermal calibration was performed, so small absolute ΔT (~0.3 °C) was treated as indeterminate and interpretation emphasized spatial patterns over single thresholds; and (iv) state that routine anaerobic culture was not performed and that microbial burden (CFU/mL) was analyzed as a continuous variable rather than semi-quantitative bins. These clarifications address the concerns about algorithm/calibration transparency, thermal reproducibility, and microbiology scope.

Revisions in the manuscript

2.2. Imaging System — end of section

…a perpendicular camera orientation to the wound surface. Fluorescence intensity was quantified using the manufacturer’s software as mean grayscale values within user-defined regions of interest (ROIs); the underlying processing algorithms are proprietary and were not accessible. In the absence of device-level calibration for microbial detection, we enhanced comparability by applying ROI-based normalization to adjacent non-fluorescent reference skin under the fixed acquisition conditions described above.

2.4. Imaging Procedure — fluorescence quantification & thermal acquisition

…A second round of fluorescence imaging was performed immediately thereafter using identical acquisition parameters.

Autofluorescent regions were identified during both sessions and documented using high-resolution still images and video recordings. The fluorescence intensity within the regions of interest was quantified using the manufacturer’s proprietary software; circular ROIs (~1 cm²) were placed on peak signal within the wound and on adjacent clinically uninvolved skin, and a relative fluorescence index (RFI; wound/reference ROI ratio) was computed to mitigate inter-frame variability. Fluorescence ROIs were co-registered between pre- and post-debridement images to maintain anatomic consistency. Multiple frames were captured per session to minimize motion or blur artifacts.

Skin emissivity was set to 0.98; ROIs (~1 cm²) were co-registered to the fluorescence maps, and temperatures were reported as ROI means. No blackbody-traceable thermal calibration was performed in this retrospective study; accordingly, small absolute temperature differences on the order of a few tenths of a degree Celsius (~0.3 °C) were treated as indeterminate, and interpretation prioritized spatial patterns (co-localization with fluorescence, perilesional extension, and—where available—serial change) over single absolute thresholds.

2.5. Microbiological Sampling and Pathogen Identification — scope & analysis

…CFU per milliliter of homogenate (CFU/mL).

Microbial burden was analyzed as a continuous variable (CFU/mL); semi-quantitative CFU categories were not assigned. Routine anaerobic culture was not performed in this retrospective cohort, which may underestimate anaerobic pathogens in chronic wounds.

2.6. Statistical Analysis — variables and tiers

Quantitative data from fluorescence imaging and microbiological cultures were analyzed to evaluate the association between the autofluorescence signal intensity and the presence of microbial infection. Fluorescence intensity was summarized using RFI, and microbial burden was treated as a continuous variable (CFU/mL). Descriptive statistics (mean, median, and standard deviation) were used to summarize the patient demographics and fluorescence measurements. Comparative analyses between fluorescence–positive- and negative-wounds were conducted using the chi-square test or Fisher’s exact test, as appropriate. Pearson’s correlation coefficient (r) was used to assess the linear relationship between the fluorescence intensity (RFI) and quantified microbial burden (CFU/mL). Diagnostic performance metrics (sensitivity, specificity, positive and negative predictive values) were calculated by comparing fluorescence results against culture as the reference standard. For descriptive reporting in Results, fluorescence intensity tiers (low, moderate, high) were used to aid presentation; all inferential analyses used continuous RFI and CFU/mL values. Statistical significance was set at p < 0.05. All statistical analyses were performed using SPSS Statistics software (version 30.0.0; IBM Corp., Armonk, NY, USA).

Comment 3

  1. Superficial Analysis of Results

 Pathogen-specific variability is poorly explained: The study reports stark differences in detection rates but provides little mechanistic insight . It attributes this to "metabolic fluorescence properties" but does not link specific pathogens to their fluorescent metabolites  or test whether wavelength adjustments could improve detection .

 Correlations ignore confounders: Fluorescence intensity correlates with exudate (r=0.72), swelling (r=0.68), and odor (r=0.68), but these associations are not adjusted for confounding variables like wound size, duration, or patient comorbidities . This limits conclusions about whether fluorescence truly reflects infection rather than unrelated wound characteristics.

 Thermal-fluorescence spatial relationships are understudied: The claim that thermal signals "extend beyond fluorescent zones" is based on qualitative observations, not quantitative analyses (e.g., frequency of this pattern or its association with subsequent infection spread) .

Response

We appreciate these points and have revised the manuscript accordingly: (i) we expanded the Discussion to explicitly link species-level detection differences to known endogenous fluorophores and wavelength considerations, citing primary sources and cross-referencing Table 7; (ii) we clarified in the Methods, Results, and Limitations that correlations between fluorescence and clinical signs are unadjusted in this preliminary cohort and therefore hypothesis-generating, and we outline a prospective plan for multivariable modeling; and (iii) we note that the observation of thermal signals extending beyond fluorescent zones was qualitative in this cohort, and we now describe a quantitative framework (e.g., predefined spatial overlap metrics) to be implemented prospectively.

Revisions in the manuscript

2.6. Statistical Analysis

For descriptive reporting in Results, fluorescence intensity tiers (low, moderate, high) were used to aid presentation; all inferential analyses used continuous RFI and CFU/mL values.

3.4. Association with Clinical Signs of Infection

The fluorescence signal intensity was positively correlated with several classical clinical signs of infection. Among them, the amount of exudate demonstrated the strongest correlation (r = 0.72; p = 0.004), followed by swelling (r = 0.68; p = 0.006), and foul odor (r = 0.68; p = 0.008). In contrast, the correlations between induration (r = 0.65, p = 0.081) and pain (r = 0.59, p = 0.137) were not statistically significant (Table 4). These correlations were calculated without covariate adjustment in this preliminary cohort and may be confounded by wound size, chronicity, or comorbid conditions; accordingly, they should be viewed as hypothesis-generating.

3.5. Thermal Imaging Findings

In several cases, thermographic signals extended beyond the borders of the fluorescence-positive zones. This peripheral thermal elevation may reflect perilesional inflammatory responses that have not yet been associated with detectable bacterial colonization. In this cohort, thermal–fluorescence relationships were assessed qualitatively; the frequency and magnitude of perilesional thermal extension will be quantified prospectively using pre-specified spatial overlap metrics (e.g., Jaccard index) and a thermal-overreach ratio. These observations highlight the complementary diagnostic utility of dual imaging. Although Fluobeam® effectively localizes microbial presence based on autofluorescence, FLIR® imaging provides additional information on the spatial extent of inflammation, potentially offering earlier insights into surrounding tissue responses.

  1. Discussion

Table 7. Species-level detectability with literature-based fluorescent metabolites

Pathogen

Classification

Detectability

Known Fluorescent Metabolites

References

Pseudomonas aeruginosa

Gram-negative aerobic rod

High

(100%)

Pyoverdine, Pyocyanin

[17,27]

Staphylococcus aureus

Gram-positive facultative anaerobic coccus

Moderate (75%)

Porphyrins
(coproporphyrin, heme biosynthetic intermediates)

[2,27]

Enterococcus faecalis

Gram-positive facultative anaerobic coccus

Low

(0%)

None identified
(minimal fluorophore production)

[18]

Klebsiella pneumoniae

Gram-negative facultative anaerobic rod

Moderate (50%)

Porphyrins
(protoporphyrin IX, coproporphyrin I)

[29]

Escherichia coli

Gram-negative facultative anaerobic rod

Moderate (50%)

Porphyrins (protoporphyrin IX)

[29]

Candida spp. (yeasts)

Fungus (yeast)

High

(100%)

Unidentified (potentially porphyrin-related fluorophores)

[2,28]

These species-level patterns are consistent with fluorescence biology reported for wound pathogens: organisms that biosynthesize strong endogenous fluorophores—such as pyoverdine in Pseudomonas aeruginosa and porphyrins in Staphylococcus aureus—tend to be more readily detected, whereas taxa with weak or absent fluorophore production (e.g., gram-positive cocci such as Enterococcus spp. and many anaerobes) show poorer detectability under our 780-nm excitation/820-nm detection protocol. Because most documented bacterial fluorophores exhibit excitation/emission peaks in the violet or visible range, extension to near-infrared excitation and detection should be interpreted with caution [14–18,28]. Moreover, the device does not perform spectral decomposition; therefore, fluorescence alone does not enable species identification. Accordingly, Table 7 is provided as an interpretive aid that links our observed detection proportions with literature-based autofluorescent metabolites and typical visible-range characteristics rather than as a species-identification key.

Limitations

Fluobeam® was used off-label for bacterial/fungal autofluorescence without microorganism-specific calibration; thermal–fluorescence co-localization was assessed qualitatively without pre-specified spatial metrics; and correlations between fluorescence and clinical signs were unadjusted and susceptible to confounding (e.g., wound size, chronicity, comorbidities). Finally, culture-based reference standards have limitations, particularly for fastidious or biofilm-associated organisms, and routine anaerobic culture was not performed in all cases.

Comment 4

  1. Overstated Conclusions

The authors conclude that combining Fluobeam® and FLIR® "improves infection detection," but no direct comparison of standalone vs. combined performance (e.g., ROC curves) is provided. Similarly, claims about "enhancing intraoperative decision-making" rely on 4 case studies, which are insufficient to demonstrate clinical utility .

 Limitations are underplayed: The study does not address how wound location (e.g., hairy vs. glabrous skin) or operator experience might affect results. It also fails to contextualize findings against existing tools (e.g., MolecuLight i:X™), leaving readers unsure of Fluobeam®’s unique value.

Response

Thank you for these thoughtful points. We have moderated our conclusions and added context. Specifically, we (i) present the four case vignettes as illustrative use scenarios rather than evidence of outcome benefit; (ii) add Table 8, a concise comparison positioning Fluobeam® + FLIR® alongside MolecuLight® and PCR, with advantages/limitations and supporting citations; and (iii) expand the Limitations to acknowledge unmeasured sources of variability (wound location—hair-bearing vs. glabrous—skin phototype, operator experience) and to note that we did not perform standalone-versus-combined ROC analyses, so the incremental benefit of the dual modality was not quantified. We also emphasize the off-label status, organism-dependent detectability, lack of spectral identification, and the need for device-level thermal calibration given the small absolute temperature differences. Finally, we revised the Conclusions to avoid overstatement, characterizing Fluobeam® as an adjunct for localizing microbial foci (not species identification) and explicitly calling for prospective, adequately powered validation.

Revisions in the manuscript

  1. Discussion

The clinical applicability of Fluobeam® was further illustrated through four representative cases, each demonstrating distinct diagnostic or surgical advantages. These cases are illustrative use scenarios (targeted debridement and sampling localization) rather than evidence of outcome benefit or clinical efficacy. 

Positioning the dual modality among adjunct tools. To place our dual-modality approach in context, we added a concise comparison of adjunct tools (Table 8). Fluobeam® + FLIR® provides rapid bedside mapping of microbial foci and perilesional spread but is off-label for bacterial detection, offers organism-dependent detectability, lacks spectral identification, and small absolute thermal deltas (~0.3 °C) warrant cautious interpretation without device-level calibration [2,33–35]. MolecuLight® is a purpose-built, portable point-of-care fluorescence platform with robust clinical evidence for bacterial-burden mapping and guidance of debridement/culture, although visibility depends on porphyrin/cyan fluorophore production and ambient-light constraints; species-level identification is not provided [12,14]. PCR delivers high analytic sensitivity/specificity and can detect targets/resistance but lacks viability confirmation, spatial mapping, and rapid turnaround compared with imaging [4]. Culture remains the clinical reference for organism identification and antimicrobial susceptibility, albeit with slower turnaround and potential false-negatives in fastidious/biofilm organisms [4]. We did not conduct head-to-head performance testing against these tools in this study; comparative accuracy and decision-impact will be evaluated prospectively. Taken together, Table 8 highlights complementary roles: imaging rapidly maps where to sample and treat, whereas laboratory tests determine what is present and how to target therapy.

Table 8. Concise comparison of adjunctive tools for wound infection assessment at the point of care and in the laboratory

Tool /
Modality

What it measures

Real-time spatial map

Species identification

Typical turnaround

Key advantages/limitations

References

Fluobeam® + FLIR®
(dual modality)

Near-infrared microbial autofluorescence (780 nm excitation / 820 nm detection) and skin surface temperature (infrared)

Yes

No

Seconds

Advantages: Real-time bedside localization of microbial foci; adjunct mapping of perilesional spread.
Limitations: Off-label for microbial detection; organism-dependent detectability; no spectral identification; small ΔT (~0.3 °C) without device-level thermal calibration; sensitive to ambient conditions and vascular status.

[2,33,34]

MolecuLight®

Porphyrins (red) and pyoverdine (cyan) under 405 nm excitation

Yes

Limited

Seconds

Advantages: Portable POC workflow; robust clinical evidence for bacterial-burden mapping; guides debridement and targeted culture.
Limitations: Detectability varies by organism (non-porphyrin/low-metabolic species less visible); ambient-light constraints; no species-level ID.

[12,14]

PCR

Microbial nucleic acids

No

Yes

Hours

Advantages: High analytic sensitivity/specificity; species/target detection; potential resistance marker genotyping.
Limitations: Cost/infrastructure needs; contamination risk; does not confirm viability; no spatial mapping; longer turnaround.

[4]

Culture

Growth of viable organisms from tissue/swab

No

Yes

Days

Advantages: Organism identification with antimicrobial susceptibility; clinical reference standard.
Limitations: Slow (days); false-negatives with fastidious/biofilm organisms; sampling error; no spatial map.

[4]

FLIR®, forward-looking infrared; POC, point-of-care; ID, species identification; NIR, near-infrared; ΔT, temperature difference; PCR, polymerase chain reaction.

Limitations

This study had several limitations. First, it was a preliminary, single-center, retrospective study with a small sample size (n=33), which limits precision and generalizability across heterogeneous wound types and microbial spectra. Inclusion was restricted to clinically suspected infections; subclinical infections and non-infectious inflammatory conditions were not systematically included, which may bias case mix and overestimate diagnostic performance relative to unselected populations. Second, to avoid optical attenuation and ensure reproducible ROI placement, we excluded wounds with excessive exudate; together with enrichment for clinically apparent infection, this may introduce selection bias. Third, although fluorescence intensity correlated with infection severity, the device lacks species-level resolution; species-level detection proportions were underpowered for several taxa (e.g., rare isolates) and are reported descriptively without inferential claims. Fourth, thermal findings were based on modest surface temperature differences (~0.3 °C) acquired without blackbody-traceable device calibration; such small deltas may fall within physiologic or measurement variability and can be influenced by environmental or vascular factors (e.g., peripheral vascular disease). We did not stratify performance by wound location (hair-bearing vs. glabrous skin), skin phototype, or operator experience, which may affect signal visibility and ROI placement, and we did not perform standalone-versus-combined receiver operating characteristic analyses, so the incremental benefit of the dual modality was not quantified. In addition, Fluobeam® was used off-label for bacterial/fungal autofluorescence without microorganism-specific calibration; thermal–fluorescence co-localization was assessed qualitatively without pre-specified spatial metrics; and correlations between fluorescence and clinical signs were unadjusted and susceptible to confounding (e.g., wound size, chronicity, comorbidities). Finally, culture-based reference standards have limitations, particularly for fastidious or biofilm-associated organisms, and routine anaerobic culture was not performed in all cases. To mitigate these limitations, we plan a prospective, adequately powered, multicenter study (target >100 participants) spanning diverse wound etiologies (burns, diabetic foot ulcers, malignancy-associated wounds), with consecutive enrollment (including subclinical infections and non-infectious mimics), pre-specified endpoints and subgroup powering, blinded image interpretation, expanded microbiology including anaerobes and molecular assays, and device calibration and repeatability assessments for fluorescence and thermal imaging.

  1. Conclusions

This study demonstrates a novel off-label use of the Fluobeam® near-infrared fluorescence system for detecting bacterial and fungal infections in chronic wounds with clinically suspected infection through autofluorescence imaging. The fluorescence signal showed meaningful correlations with microbial burden, clinical signs, and thermal imaging patterns, indicating its potential as a real-time, noninvasive diagnostic adjunct for localizing microbial foci rather than species identification. Combined use with FLIR® thermal imaging provided complementary information on host response and spatial extent of infection severity. These findings support the integration of fluorescence-based imaging in routine wound care, while recognizing species-dependent detectability and modest absolute thermal differences. However, prospective, adequately powered studies are required to validate its performance in broader clinical settings and pathogen spectra.

Reviewer 3 Report

Comments and Suggestions for Authors

The manuscript you submitted meets the criteria of being novel, useful, and easily accessible regarding wound assessment. My concern is whether this type of approach to diagnosing infected or non-infected wounds would be more useful, taking into account the progression of the wound itself. It would be very instructive to include cases with before and after effective or ineffective treatment. How do you support this type of argument? 

Author Response

Response to Reviewer 3

The manuscript you submitted meets the criteria of being novel, useful, and easily accessible regarding wound assessment. My concern is whether this type of approach to diagnosing infected or non-infected wounds would be more useful, taking into account the progression of the wound itself. It would be very instructive to include cases with before and after effective or ineffective treatment. How do you support this type of argument? 

Response

Thank you for this constructive suggestion. We added longitudinal before/after images for three cases and a concise paragraph in the Discussion to illustrate wound trajectory. Figure 5 now shows: (A) a 3-month image of Case 1 with a small residual slough at the prior fluorescence-positive margin (marked by a yellow dashed ellipse), which prompted additional targeted debridement and subsequently progressed to complete epithelialization; and (B–C) durable 12-month epithelialization for Cases 2 and 3. We explicitly state that these images are illustrative use scenarios rather than proof of outcome benefit and note that standardized serial fluorescence and thermal acquisitions were not available in this retrospective cohort. We will address this prospectively with predefined outcome endpoints and serial imaging.

Revisions in the manuscript

  1. Discussion

Figure 5 illustrates longitudinal outcomes for three cases. Panel A shows the 3-month status of Case 1 with partial epithelialization and a small residual slough at the same peripheral margin that corresponded to the baseline fluorescence-positive hotspot, indicated by a yellow dashed ellipse. During follow-up this region underwent targeted debridement and then progressed to complete epithelialization. Panels B and C show the 12-month outcomes of Cases 2 and 3 with durable epithelialization. These images are presented as illustrative use scenarios rather than evidence of outcome benefit, and standardized serial fluorescence and thermal acquisitions were not available in this retrospective cohort.

Figure 5. Long-term outcomes. (A) Case 1 at 3 months: residual slough at the prior fluorescence-positive margin (yellow dashed ellipse), subsequently debrided and healed. (B) Case 2 at 12 months: durable epithelialization. (C) Case 3 at 12 months: durable epithelialization.

Limitations

Future work will include serial, calibrated Fluobeam®/FLIR® acquisitions during long-term follow-up with blinded readers to test reproducibility and clinical relevance of longitudinal signal changes.

Round 2

Reviewer 1 Report

Comments and Suggestions for Authors

Considering the amendments made, it seems that the current draft, in its present form, satisfies the criteria for publication in this journal.